# Balancing Bias in Two-sided Markets for Fair Stable Matchings

**Siyuan Wu**[*]
University of Macau
asuka.wsy@connect.umac.mo

**Leong Hou U**
University of Macau
ryanlhu@um.edu.mo

**Panagiotis Karras**
U. of Copenhagen & Aarhus U.
piekarras@gmail.com

## Abstract

The Balanced Stable Marriage (BSM) problem aims to find a stable matching in a *two-sided* market that minimizes the *maximum dissatisfaction* among two sides. The classical Deferred Acceptance algorithm merely produces an unfair stable marriage, providing optimal partners for one side while partially assigning pessimal partners to the other. Solving BSM is NP-hard, thwarting attempts to resolve the problem exactly. As the instance size increases in practice, recent studies have explored heuristics for finding a fair stable marriage but have not found an *exact* optimal solution for BSM efficiently. Nevertheless, in this paper we propose an efficient algorithm, Isorropia, that returns the *exact* optimal solution to practical BSM problem instances. Isorropia constructs two sets of candidate rotations from which it builds three sets of promising antichains, and performs local search on those three sets of promising antichains. Our extensive experimental study shows that Isorropia surpasses the time-efficiency of baselines that return the *exact* solution by up to *three orders of magnitude*.

## 1 Introduction

Given a two-sided market, where each agent (conventionally, man or woman) ranks those on the other side by a strict order (Roth, 1984), the *stable marriage problem* (SMP) (Gale & Shapley, 1962) seeks a *stable matching* between the two sides, such that no pair of agents in separate matchings would both rather be matched with each other than with their assigned matches. SMP widely finds real applications involving two-sided markets, such as assigning residents to hospitals (Gusfield & Irving, 1989; Askalidis et al., 2013), students to universities (Gale & Shapley, 1962; Teo et al., 2001; Baïou & Balinski, 2004; Saif et al., 2020), reviewers to papers (Long et al., 2013; Kou et al., 2015), and jobseekers to jobs (Roth, 1984; Liebowitz & Simien, 2005).

Considering fairness in stable marriage problems, the celebrated *Deferred Acceptance* (DA) algorithm (Gale & Shapley, 1962), offered an allocation *optimal* for each agent on the one side and *pessimal* for each one on the other in $O(n^2)$ time (McVitie & Wilson, 1971; Irving & Leather, 1986). Since then, several stable marriage fairness objectives have been suggested (Gusfield & Irving, 1989; Iwama & Miyazaki, 2008). The *regret cost* objective (Gusfield, 1987) calls to minimize the dissatisfaction of the worst-off individual among all agents, while it only caters for fairness among individual agents, but not among sides. The *egalitarian cost* aims to minimize the sum of all agents' dissatisfaction, but not the gap between two sides, which overly gratifies the preferences of one side. The *sex-equality cost* (Kato, 1993) measures the gap among the two sides' collective dissatisfaction, yet therefore penalizes solutions in which both sides would be better off though the difference among them would be higher. Most consequentially, the *balance cost* (Feder, 1992; Gupta et al., 2021) aims to minimize the highest collective dissatisfaction among the two sides, raising the *Balanced Stable Marriage* problem (BSM). Contrariwise, the BSM objective considers the incentives of both sides in a balanced manner, endorsing a decrease in collective dissatisfaction, which renders both sides better off even at the expense of fairness among the two.

Unfortunately, BSM is hard and calls for algorithms that efficiently explore the solution space (Irving, 2016; Roth, 2018; Dworczak, 2021). Technically, the minimization of the *balance cost* objective is

---
[*]The work was done partially while visiting Aarhus University.

an NP-hard problem (Kato, 1993; Feder, 1995; Gupta et al., 2021). Pragmatically, the set of possible stable matchings can grow large in practice (Hassidim et al., 2017), while the problem instance size is also usually large. In China, over 10 million students apply for admission to higher education annually via a centralized process (Manlove, 2013). Similar schemes, in which students are ranked by some score (Biró & Kiselgof, 2015), are used in several national schemes (Braun et al., 2010; Biró, 2008; Romero-Medina, 1998; Balinski & Sönmez, 1999; Biró & Kiselgof, 2015; Ágoston et al., 2016) and US school districts (Abdulkadiroğlu & Sönmez, 2003; Abdulkadiroğlu et al., 2005b;a). Even when the problem instance size is small, the number of all stable matchings grows exponentially in the worst case (Knuth, 1976; 1997). For some hard instances, the exact maximum number of stable matchings an instance can have is known to be at least $2.28^n$ and at most $c^n$ for some universal constant $c$ (Karlin et al., 2018). Mostly heuristic methods try to find a 'fair' stable marriage without seeking a specific objective score but using unbiased proposal and acceptance strategies to search the solution space (Viet et al., 2016b;a; Dworczak, 2016; 2021; Tziavelis et al., 2019). Tziavelis et al. (2019) propose a local-search heuristic, HybridMultiSearch (HMS), that finds a stable marriage with high equity in quadratic time, yet provide no guarantees with respect to any objective. As detailed in Section 2, no previous study *finds the exact solution to BSM in practice*, while existing heuristics focus on devising proposal-and-acceptance strategies.

In this paper, we propose ISORROPIA,[1] an algorithm that effectively finds *exact* balanced stable marriages in practical problem instances, via efficient local search and intensive pruning of the solution space. The structure of all feasible stable marriages in a problem instance can be compactly represented by a partially ordered set of *rotation* structures (Gusfield & Irving, 1989; Irving & Leather, 1986), viewed as a directed acyclic graph (DAG), the *rotation graph*. Each stable marriage corresponds to a set of rotations known as *antichain*. Contrariwise to previous heuristics, our solution performs local search in a subset of promising antichains in the rotation graph. To facilitate this search, we delimit and search three sets of promising antichains, built from two sets of candidate rotations, by exploiting locally optimal constructs of the dissatisfaction function under a domination relationship among stable marriages. Notably, ISORROPIA *finds the exact solution, despite using local search*. We extend ISORROPIA to find the exact sex-equal stable marriage and show that it surpasses the time-efficiency of baselines that return the *exact* solution by up to *three orders of magnitude*.

## 2 RELATED WORK

**Fairness objectives and tractability.** The Gale-Shapley algorithm (1962) finds a one-side optimal stable marriage in $O(n^2)$ time, trading the satisfaction of one side in favor of the other. Several problem variants define different objectives, such as regret cost (Gusfield, 1987), egalitarian cost (Irving et al., 1987), and sex-equality cost (Kato, 1993). Feder (1995) proves that BSM is NP-complete and Gupta et al. (2021) give a parameterized complexity analysis. Let $P_m$ ($P_w$) denote the individual dissatisfaction of a man $m$ (woman $w$) and $C_M$ (or $C_W$) the accumulated dissatisfaction of men (women). $P_a$ represents the individual dissatisfaction of an agent (i.e, man or woman) and $\mathcal{U} = \{\mu_1, \mu_2, \ldots\}$ the space of all stable marriages. Table 1 summarizes the fairness objectives.

BSM is approximable within a factor of 2 in $O(\upsilon \log(\omega^2/\upsilon + 2))$ time (Feder, 1995), where $\upsilon$ is the number of clauses and $\omega$ is the explicit width in the related balanced 2SAT problem. From the viewpoint of parameterized complexity, a prior work (Gupta et al., 2021) gives two parameterizations of BSM by two versions of the parameter $t$, i.e., $t = k - \min\{C_M, C_W\}$ and $t = k - \max\{C_M, C_W\}$) such that the balance cost is not larger than $k$, where the first one has an FPT algorithm and another one is W[1]-hard. Contrariwise, we find the exact solution to BSM in a manner that performs efficiently in practical problem instances.

**Existing methods** that find a fair stable marriages (Table 1) follow one of these orientations:

- *Proposal algorithms.* These algorithms adopt a procedure similar to the Gale-Shapley algorithm (1962), letting agents reach a stable matching by exchanging, accepting, and rejecting proposals across the two sides. The *randomized order mechanism* (ROM) generates a finite chain of matchings that terminates at a stable matching. In each iteration, it randomly introduces an individual (Ma, 1996) or a pair (Roth & Sotomayor, 1990) into the proposal procedure. However, ROM cannot enumerate all stable marriages and is inherently biased in favor of each randomly selected

---

[1]From Greek ἰσορροπία, balance, equipoise'.

Table 1: Fairness objectives in the Stable Marriage (SM) Problem

| Problem | Objective | Tractability |
|---|---|---|
| Minimum Regret SM (Gusfield, 1987) | $\min\limits_{\mu \in \mathcal{U}} \max\limits_{\langle m,w \rangle \in \mu} \max\left\{P_m(w), P_w(m)\right\}$ | $O(n^2)$ |
| Egalitarian SM (Irving et al., 1987) | $\min\limits_{\mu \in \mathcal{U}} C_M(\mu) + C_W(\mu)$ | $O(n^4)$ |
| Sex-equal SM (Kato, 1993; Iwama et al., 2010) | $\min\limits_{\mu \in \mathbb{M}} \lvert C_M(\mu) - C_W(\mu) \rvert$ | NP-hard |
| Balanced SM (Feder, 1992; Gupta et al., 2021) | $\min\limits_{\mu \in \mathcal{U}} \max\{C_M(\mu), C_W(\mu)\}$ | NP-hard |

individual (Tziavelis et al., 2020). Other works have proposed alternative orders of proposals to enhance fairness via an unbiased treatment of the two sides. EROM (Romero-Medina, 2005) lets agents propose to each other with progressive receptiveness. SWING (Everaere et al., 2013) and ESMA (Giannakopoulos et al., 2015) let all individuals re-propose to others in iterations. Deferred Acceptance with Compensation Chains (DACC) (Dworczak, 2016; 2021) reaches a practically fair stable matching by compensating abandoned agents in $O(n^4)$ time, while PowerBalance (Tziavelis et al., 2019) finds a fair stable matching in $O(n^2)$ by using a *stricter* proposal acceptance criterion.

- *Linear programming and satisfiability.* The matching problem with stability constraints can be formulated as a linear programming problem under a set of linear inequality constraints (Rothblum, 1992) and by a SAT formula (Siala & O'Sullivan, 2017). LOTTO (Aldershof et al., 1999) follows a similar formulation, eliminating redundant constraints in each iteration and assigning a random agent to its best available preference, hence also exhibits bias (Tziavelis et al., 2019).

- *Local search on the stable marriage lattice.* In any SMP instance, the set of all possible stable marriages forms a distributive lattice (Irving & Leather, 1986). SML2 (Gelain et al., 2013) starts from a random matching and iteratively eliminates selected blocking pairs (i.e., pairs of agents who would rather be matched with each other than with their assigned matches) to transform it to a stable one by *local search* on the lattice. BiLS (Viet et al., 2016b;a) performs local search on the lattice with a greedy strategy and a probability for random movement. Nevertheless, these empirical methods are constrained by the size of stable matching lattice, which can grow up to exponential size in $n$ (Irving & Leather, 1986), while local search may get stuck in local optima.

- *Rotation-based model.* While computing and storing the distributive lattice structure may be unattainable, a more compact structure, the *rotations poset*, i.e., a directed acyclic graph organizing *rotations* (i.e., sub-matchings) (Irving & Leather, 1986; Gusfield & Irving, 1989) also represents all stable solutions. To minimize *egalitarian cost*, it suffices to find a minimum cut on this graph in $O(n^4)$ time (Gusfield & Irving, 1989). A recent work (Bozec-Chiffoleau et al., 2024) solves the robust stable marriage problem via rotation-based model, which reduces the search space and speed up the exploration on rotation graph. Nevertheless, some algorithms based on the rotation graph are unclear. For example, an algorithm for sex-equal stable marriages by Romero-Medina (2001) requires finding rotations that change signs, without suggesting an implementation.

Further, some *hybrid methods* embody more than one of these orientations. Deferred Local Search (DLS) and HybridMultiSearch (HMS) generate additional fair stable marriages by pursuing local search strategies starting from the output of PowerBalance (Tziavelis et al., 2019). However, these algorithms are mostly heuristics, aiming for *procedural fairness* without targeting a specific equity cost measure (e.g., sex-equality cost or balance cost). Contrariwise, our algorithm efficiently finds the *exact* balanced stable marriage via local search on rotation graph.

## 3 PROBLEM STATEMENT AND PRELIMINARIES

A stable marriage instance is defined as $\mathcal{I} = (M, W, P)$, where $M$ and $W$ are the two sets of agents (conventionally exemplified as men and women) of the same size $n$, and $P$ is a set of $2n$ *preference lists*, list $P_i$ for agent $i$, which rank in descending order those on the other side. An example instance is provided in Appendix A.1. $P_m(w)$ denotes the position of $w$ in $P_m$ and $P_w(m)$ that of $m$ in $P_w$. In effect, $P_i(j)$ also expresses the extent of $i$'s *dissatisfaction* with $j$. A matching $\mu$ has $n$ disjoint $\langle m, w \rangle$ pairs. We use $\mu(m) = w$ and $\mu(w) = m$ to denote that $\langle m, w \rangle$ is a pair in

matching $\mu$. If $m$ prefers $w$ to $\mu(m)$ and $w$ prefers $m$ to $\mu(w)$, then $\langle w, m \rangle$ is *blocking pair* in $\mu$. A *stable marriage* is a matching without blocking pairs.

**Balanced Fairness.** Given a stable matching $\mu$, let $C_M(\mu)$ and $C_W(\mu)$ represent the sum of preferences for the assigned matches on the two sides:

$$C_M(\mu) = \sum_{\langle m,w \rangle \in \mu} P_m(w), \quad C_W(\mu) = \sum_{\langle m,w \rangle \in \mu} P_w(m) \tag{1}$$

$C_M(\mu)$ and $C_W(\mu)$ reflect the cumulative dissatisfaction of all agents on the $M$-side and $W$-side, respectively. The balanced stable marriage problem (Gupta et al., 2021) aims to find a stable marriage $\mu^*$ that minimizes the worst dissatisfaction of the disadvantage side:

$$C(\mu^*) = \min_{\mu \in \mathcal{U}} \max \{ C_M(\mu), C_W(\mu) \} \tag{2}$$

where $\mathcal{U} = \{ \mu_1, \mu_2, \ldots, \mu_N \}$ is the set of all stable marriages, whose size $N$ is exponential in the worst case (Irving & Leather, 1986).

To facilitate the discussion, we introduce the function $\mathsf{Worse}(\mu)$, which determines the disadvantaged side in a stable marriage $\mu$ and hence yields the balance cost:

$$\mathsf{Worse}(\mu) = \begin{cases} W\text{-side} & \text{if } C_M(\mu) \leq C_W(\mu) \\ M\text{-side} & \text{if } C_M(\mu) > C_W(\mu) \end{cases} \tag{3}$$

**The Structure of All Stable Marriages.** Given a stable marriage instance $\mathcal{I}$, all stable marriages $\mathcal{U}$ are composed of (1) two side-pessimal stable marriages ($\mu_W$ and $\mu_M$) and (2) other stable marriages. First, two side-pessimal stable marriages can be generated by *Deferred Acceptance* (DA) algorithm upon its first termination. Then, other stable marriages can be generated by re-assigning some pairs from $\mu_W$, and finally it can reach at $\mu_M$. The re-assignment follows a set of DA procedures (i.e., break stable marriages and apply DA algorithm for multiple times), which can be compactly represented by a set of rotation nodes.[2]

The *Deferred Acceptance* (DA) algorithm (Gale & Shapley, 1962) lets each man $m$ start from the first preference and sequentially propose to the next most preferable woman in the order of $P_m$, as long as the man finds itself being single. Each woman $w$ accepts a $(m, w)$ proposal if the woman is single or prefers $m$ to the current partner $\mu(w)$. The DA algorithm (Gale & Shapley, 1962) outputs a stable marriage optimal for each agent on one side and pessimal for each agent on the other side (McVitie & Wilson, 1971; Irving & Leather, 1986), i.e., we get $\mu_0$ if men propose to women and we get $\mu_4$ if women propose to men. As shown in Table 2, we denote these two outputs as $\mu_W$ and $\mu_M$, where the subscript denotes the side that gets a pessimal outcome.

A stable marriage $\mu$ *dominates* another $\mu'$, or $\mu \prec \mu'$, if each agent on the $M$-side gets a no less preferable partner in $\mu$ than in $\mu'$, and, as stability implies, each agent on the $W$-side gets a no more preferable partner in $\mu$ than in $\mu'$. The set of all stable marriages forms a distributive lattice (Gusfield & Irving, 1989), where $\mu_W$ dominates, and $\mu_M$ is dominated by, any other stable marriage.

To compactly represent the breakable pairs and the corresponding re-assigned pairs for each DA process, we use the construct of *rotation* (Irving, 1985; Irving & Leather, 1986). A rotation belonging to (or *exposed in*) $\mu$ is an ordered sub-list of matched pairs $r = \{ \langle m_i, \mu(m_i) \rangle, \langle m_{i+1}, \mu(m_{i+1}) \rangle, \ldots, \langle m_{i+d}, \mu(m_{i+d}) \rangle \}$. Given a $\mu$ that exposes a rotation $r$, we can break the marriage of $m_i$ in rotation $r$ and apply the DA algorithm to let $m_i$ propose to the next most preferable woman, eventually being assigned with woman $\mu(m_{i+1})$ who abandons man $m_{i+1}$; likewise, $m_{i+1}$ will then be matched with $\mu(m_{i+2})$, and so on until we reach $\mu(m_i)$ in full cycle. Intuitively, each of the men $m_i, m_{i+1}, \ldots, m_{i+d}$ is matched to a woman less preferable to him, $\mu(m_{i+1}), \mu(m_{i+2}), \ldots, \mu(m_i)$ respectively, while each of the woman $\mu(m_i), \mu(m_{i+1}), \ldots, \mu(m_{i+d})$ is matched to a man more preferable to the woman, $m_{i+d}, m_i, \ldots, m_{i+d-1}$ respectively. Thus, the ensuing matching $\mu'$ is still stable. We call this re-coupling *rotation elimination*, denoted as $\mu/r \to \mu'$. By *eliminating* the rotation $r$, we can obtain a new stable marriage $\mu'$. Certainly, $\mu$ dominates $\mu'$.

---

[2] For more details, readers can refer to (Gusfield & Irving, 1989; Irving & Leather, 1986) and Appendix A.1.

Table 2: Rotation elimination and balance costs for all stable marriages

| $\mu$ | Matches | | | | | $a$ | $s$ | $C_M(\mu)$ | $C_W(\mu)$ | Worse($\mu$) | Balance Cost |
|---|---|---|---|---|---|---|---|---|---|---|---|
| $\mu_0(\mu_W)$ | $\langle m_1,w_1\rangle$ | $\langle m_2,w_5\rangle\langle m_3,w_3\rangle\langle m_4,w_4\rangle$ | $\langle m_5,w_2\rangle$ | | | $\emptyset$ | $\emptyset$ | 9 | 18 | $W$-side | 18 |
| $\mu_1$ | $\langle m_1,w_2\rangle$ | $\langle m_2,w_5\rangle\langle m_3,w_3\rangle\langle m_4,w_4\rangle$ | $\langle m_5,w_1\rangle$ | | | $\{r_1\}$ | $\{r_1\}$ | 11 | 16 | $W$-side | 16 |
| $\mu_2$ | $\langle m_1,w_1\rangle$ | $\langle m_2,w_3\rangle\langle m_3,w_4\rangle\langle m_4,w_5\rangle$ | $\langle m_5,w_2\rangle$ | | | $\{r_2\}$ | $\{r_2\}$ | 12 | 11 | $M$-side | 12 |
| $\mu_3$ | $\langle m_1,w_2\rangle$ | $\langle m_2,w_3\rangle\langle m_3,w_4\rangle\langle m_4,w_5\rangle$ | $\langle m_5,w_1\rangle$ | | | $\{r_1,r_2\}$ | $\{r_1,r_2\}$ | 14 | 9 | $M$-side | 14 |
| $\mu_4(\mu_M)$ | $\langle m_1,w_2\rangle$ | $\langle m_2,w_3\rangle\langle m_3,w_4\rangle$ | $\langle m_4,w_1\rangle\langle m_5,w_5\rangle$ | | | $\{r_3\}$ | $\{r_1,r_2,r_3\}$ | 17 | 6 | $M$-side | 17 |

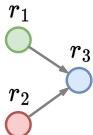

Figure 1: Rotation Graph $G$

The set of all rotations $R$ forms a *partially ordered set* (poset) $(R, \rightarrow)$ (Irving & Leather, 1986). A partial order $r \rightarrow r'$ indicates that $r'$ is only exposed *after* eliminating $r$. The poset $(R, \rightarrow)$ is denoted by a directed acyclic graph $G = (R, E)$, where nodes stand for rotations and an edge $(r, r')$ for a *direct* partial order $r \rightarrow r'$, i.e., $\nexists r_*|r \rightarrow r_* \rightarrow r'$, guaranteeing connectivity. Figure 1 shows an example of a rotation graph $G$. We denote the predecessors of $r$ as $Pred(r)$, i.e., for each $r' \in Pred(r)$, $r$ is reachable from $r'$. Finding all rotations and edges in $G$ costs $O(n^2)$ time (Irving et al., 1987; Gusfield, 1987); this construction is a preprocessing step beneath the core of our problem.

An *antichain* $a$ is a subset of $R$ such that no rotation in $a$ is a predecessor of another. Given an antichain $a$, we can construct a unique *closed subset* $s = \bigcup_{r \in a}\{r\} \cup Pred(r)$, which contains all rotations in $a$ and their predecessors. For $a = \{r_3\}$ in Figure 1, we should eliminate rotations in its closed subset, $s = \{r_1, r_2, r_3\}$, according to partial order relationship starting from $\mu_W$, so that each rotation is exposed in the elimination process.

Let $\mathcal{A}$ be the set of antichains and $\mathcal{S}$ the set of closed subsets in $G$. Theorem 1 (Irving & Leather, 1986) provides a foundational fact on the structure of all stable marriages.

**Theorem 1** (Relationship between antichains, closed subsets, and stable marriages). *(Irving & Leather, 1986) In any stable marriage instance there is a one-to-one relationship among antichains $\mathcal{A}$, closed subsets $\mathcal{S}$ and stable marriages $\mathcal{U}$. Enumerating all stable marriages is #P-complete.*

In other words, for any antichain $a$, we can find a corresponding closed subset $s$ and stable marriage $\mu$ via rotation elimination $\mu_W/s \rightarrow \mu_a$. We refer to these concepts (i.e., $\mu$, $a$, and $s$) interchangeably without loss of clarity. As all stable marriages listed in Table 2, the balanced stable marriage is $\mu_2$.

## 4 LOCAL SEARCH ALGORITHM

In this section, we introduce our algorithm, ISORROPIA, which efficiently returns the exact solution to the BSM problem by locally searching *three sets of promising antichains*, $\mathcal{A}_{\text{I}}, \mathcal{A}_{\text{II}}, \mathcal{A}_{\text{III}}$ build from two sets of candidate rotations, $R_{\triangleleft}, R_{\triangleright}$.

### 4.1 LOCAL OPTIMALITY

To find the balanced stable marriage $\mu^*$ (cf. Equation 2), we exploit four properties of the variation of $C_M$ and $C_W$ along the rotation elimination process from the extreme $\mu_W$ to $\mu_M$. As rotation elimination degrades the matches of the $M$-side and upgrades the matches on the $W$-side (Irving & Leather, 1986; Gusfield, 1987), the following monotonicity property follows.

**Property 1.** *Starting from $\mu_W$, a rotation elimination $\mu/r \rightarrow \mu'$, increases $C_M$ and decreases $C_W$, i.e., $C_M(\mu) < C_M(\mu')$ and $C_W(\mu) > C_W(\mu')$.*

**Remark.** *Given a rotation $r$, the rotation elimination results in that each agent $m$ in $r$ gets a less preferable partner and each agent $w$ in $r$ gets a more preferable partner (see Section 3), bringing to the strict increase of $C_M$ and the strict decrease of $C_W$ respectively by Equation 1.*

When eliminating a set of rotations, $\mu/R_* \rightarrow \mu'$, for each man $m \in r$ and $r \in R_*$, $\mu(m)$ is a better partner than $\mu'(m)$ to the agent $m$, while other agents in $M$ follow that $\mu'(m) = \mu(m)$. In effect, the resulting matching $\mu'$ is dominated by $\mu$ (i.e., $\mu \prec \mu'$). From Property 1 we derive Property 2, which determines the worse-off side of one stable marriage from the worse-off side of another stable marriage via the domination relationship.

**Property 2.** *For $\mu \prec \mu'$, Worse($\mu$) = $M$-side $\Rightarrow$ Worse($\mu'$) = $M$-side and Worse($\mu'$) = $W$-side $\Rightarrow$ Worse($\mu$) = $W$-side.*

**Remark.** *The property holds because $\mu \prec \mu'$ implies that $C_W(\mu') < C_W(\mu)$ and $C_M(\mu) < C_M(\mu')$, while $\mathsf{Worse}(\mu) = M$-side implies that $C_W(\mu) < C_M(\mu)$, hence $C_W(\mu') < C_W(\mu) < C_M(\mu) < C_M(\mu')$, therefore $\mathsf{Worse}(\mu') = M$-side. The implication from $\mathsf{Worse}(\mu') = W$-side follows by analogous reasoning.*

The *local optimality* properties follow from Property 2.

**Property 3.1.** *If $\mu \prec \mu'$, $\mathsf{Worse}(\mu') = W$-side $\Rightarrow C_W(\mu') < C_W(\mu)$ (i.e., $\mu'$ is better).*

**Property 3.2.** *If $\mu \prec \mu'$, $\mathsf{Worse}(\mu) = M$-side $\Rightarrow C_M(\mu) < C_M(\mu')$ (i.e., $\mu$ is better).*

**Remark.** *If $\mu \prec \mu'$, it follows that both $\mu$ and $\mu'$ have $W$-side as the disadvantaged side (Property 2) and $C_W(\mu') < C_W(\mu)$ (Property 1), where $\mu'$ has a better balance cost as Property 3.1. The implication from Property 3.2 follows by analogous reasoning.*

## 4.2 MAIN IDEA

Given a rotation $r$, the $r$-related antichain contains only $r$, $a_r = \{r\}$, while the $r$-related closed subset is $s_r = \{r\} \cup Pred(r)$. We derive the $r$-related stable marriage $\mu_r$ by rotation elimination as $\mu_W/s_r \rightarrow \mu_r$. We divide all rotations in two subsets based on the side of the market on which their $r$-related stable marriages are *disadvantaged*, i.e., the value of $\mathsf{Worse}(\mu_r)$ by Equation 3.

**Definition 1** (Side-Disadvantaged rotations, $R_M$ and $R_W$). *The set of rotations disadvantaged on the $M$-side is the set $R_M = \{r | \mathsf{Worse}(\mu_r) = M$-side$\}$ and the set of rotations disadvantaged on the $W$-side is the set $R_W = \{r | \mathsf{Worse}(\mu_r) = W$-side$\}$.*

Clearly, it is $R_M \cup R_W = R$ and $R_M \cap R_W = \emptyset$. Since an antichain is a set of rotations, we distinguish three disjoint sets of antichains: (i) Antichains $a \in \mathcal{A}_M$ that $a$ only contains rotation(s) in $R_M$; (ii) Antichains $a \in \mathcal{A}_W$ that $a$ only contains rotation(s) in $R_W$; (iii) Antichains $a \in \mathcal{A}_{MW}$ that $a$ contains rotations in both $R_M$ and $R_W$.

Figure 2 depicts the disjoint sets of the three types of antichains, where $\mathcal{A}_M \cup \mathcal{A}_W \cup \mathcal{A}_{MW} = \mathcal{A}$.[3]

Since each stable marriage can be generated by its corresponding antichain (Theorem 1), the matching problem is transformed into a graph searching problem that finds a set of rotations. Our algorithm, ISORROPIA, reduces the search space by extracting and locally searching three sets of promising antichains, namely $\mathcal{A}_{\mathrm{I}} \subset \mathcal{A}_W$, $\mathcal{A}_{\mathrm{II}} \subset \mathcal{A}_W$ and $\mathcal{A}_{\mathrm{III}} \subset \mathcal{A}_M$, while discarding $\mathcal{A}_{MW}$.

## 4.3 MIN-MAX OPTIMIZATION

The BSM problems calls for a min-max optimization where the $\max$ operator in Equation 2 leads to a non-convex objective function. To render the objective more manageable, we drop the $\max$ operator in Equation 2, splitting it in two cases as follows:

$$\mu^* = \arg\min_{\mu \in \mathcal{U}} \begin{cases} C_W(\mu) & \text{if } \mathsf{Worse}(\mu) = W\text{-side} \\ C_M(\mu) & \text{if } \mathsf{Worse}(\mu) = M\text{-side} \end{cases} \tag{4}$$

Thereby, we aim to find the minimum $C_W(\mu)$ when $C_M(\mu) \leq C_W(\mu)$ and the minimum $C_M(\mu)$ when $C_M(\mu) > C_W(\mu)$, and return the stable marriage having the least score among these two. Building upon it, We search the rotation graph $G$ while minimizing the two possible manifestations of the balance cost, that is, $C_M(\mu)$ for $\mathsf{Worse}(\mu) = M$-side and $C_W(\mu)$ for $\mathsf{Worse}(\mu) = W$-side. Unfortunately, $G$ is not neatly divided in two subgraphs such that $\mathsf{Worse}(\mu) = M$-side in one and $\mathsf{Worse}(\mu) = W$-side in the other. As each stable matching corresponds to a closed subset (i.e., combination) of rotations, a rotation $r$ exposed in one stable matching $\mu$ with $\mathsf{Worse}(\mu) = W$-side may also be exposed in another stable marriage $\mu'$ with $\mathsf{Worse}(\mu') = M$-side. For example in Table 5, both $a_{r_5}$ and $a_y$ have $r_5$ while their worse-off sides are different. We achieve the above purpose by exploring three sets of promising antichains, $\mathcal{A}_{\mathrm{I}}$, $\mathcal{A}_{\mathrm{II}}$, and $\mathcal{A}_{\mathrm{III}}$ articulated in Table 4.

$\mathcal{A}_{\mathrm{I}}$ **and** $\mathcal{A}_{\mathrm{II}}$. First, we consider antichains $a \in \mathcal{A}_W$, which only contain rotations in $R_W$ (pink area in Figure 2). We partition antichains in $\mathcal{A}_W$ in two blocks depending on the disadvantaged side in their corresponding stable marriages $\mu_a$:

---

[3] It should be $\mathcal{A}_M \cup \mathcal{A}_W \cup \mathcal{A}_{MW} \cup \{\emptyset\} = \mathcal{A}$. In ISORROPIA, we have to check two extreme cases of $\mu_M$ and $\mu_W$, corresponding to $s = \emptyset$ and $s = R$, hence can ignore these two corresponding antichains in $\mathcal{A}$.

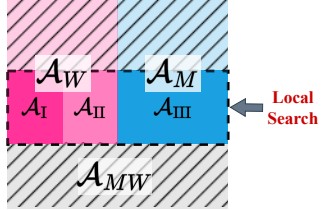

Figure 2: We first divide the search space $\mathcal{A}$ into $\mathcal{A}_M$ (blue area), $\mathcal{A}_W$ (pink area) and $\mathcal{A}_{MW}$ (gray area). ISORROPIA only locally searches $\mathcal{A}_I$, $\mathcal{A}_{II}$ and $\mathcal{A}_{III}$ (dotted box).

Table 3: The exact solution can be found in the local search space by Theorem 2, 3 and 4. For each theorem, an antichain in unpromising antichains cannot yield a better result in terms of balance cost than the optimal result found in promising antichains.

| Theorem | Promising Antichains | Unpromising Antichains |
|---|---|---|
| Theorem 2 | $\mathcal{A}_I \cup \mathcal{A}_{II}$ | $\mathcal{A}_W \setminus (\mathcal{A}_I \cup \mathcal{A}_{II} \cup \{a_{\ell_W}\})$ |
| Theorem 3 | $\mathcal{A}_{III}$ | $\mathcal{A}_M \setminus \mathcal{A}_{III}$ |
| Theorem 4 | $\mathcal{A}_{III}$ | $\mathcal{A}_{MW}$ |

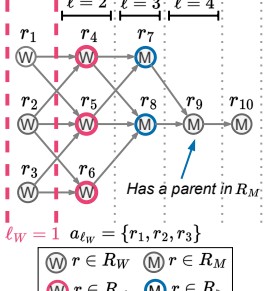

Figure 3: $R_\triangleleft$ and $R_\triangleright$

Table 4: Three sets of promising antichains

| Promising Antichains | Contained Rotations | Worse($\mu_a$) |
|---|---|---|
| $\mathcal{A}_I$ | $a = \{r \mid \forall r \in R_W \wedge \exists r \in R_\triangleleft\}$ | $W$-side |
| $\mathcal{A}_{II}$ | $a = \{r \mid \forall r \in R_W \wedge \exists r \in R_\triangleleft\}$ | $M$-side |
| $\mathcal{A}_{III}$ | $a = \{r \mid r \in R_\triangleright\}, |a| = 1$ | $M$-side |

Table 5: Examples of stable marriages.

| $\mu$ | $a$ | $s$ | $C_M(\mu)$ | $C_W(\mu)$ | Worse($\mu$) | Balance Cost |
|---|---|---|---|---|---|---|
| $\mu_{r_4}$ | $\{r_4\}$ | $\{r_1, r_2, r_4\}$ | 19 | 35 | $W$-side | 35 |
| $\mu_x$ | $\{r_3, r_4\}$ | $\{r_1, \cdots, r_4\}$ | 25 | 31 | $W$-side | 31 |
| $\mu_{r_5}$ | $\{r_5\}$ | $\{r_1, r_2, r_3, r_5\}$ | 26 | 30 | $W$-side | 30 |
| $\mu_y$ | $\{r_4, r_5\}$ | $\{r_1, \cdots, r_5\}$ | 28 | 26 | $M$-side | 28 |
| $\mu_{r_7}$ | $\{r_7\}$ | $\{r_1, \cdots, r_5, r_7\}$ | 33 | 24 | $M$-side | 33 |

- $\{a \in \mathcal{A}_W \mid \text{Worse}(\mu_a) = W\text{-side}\}$; Some antichains in $\mathcal{A}_W$ derives $W$-disadvantaged stable marriages. For example, in Figure 3 and Table 5, both $r_3$ and $r_4$ are $W$-disadvantaged rotations. $\mu_{r_4}$ is disadvantaged on $W$-side, and $\mu_x$, corresponding to antichain $\{r_3, r_4\}$, also derives a $W$-disadvantaged stable marriage.

- $\{a \in \mathcal{A}_W \mid \text{Worse}(\mu_a) = M\text{-side}\}$; Some antichains in $\mathcal{A}_W$ derives $M$-disadvantaged stable marriages [4]. For example, both $r_4$ and $r_5$ are in $R_W$. $\mu_{r_4}$ and $\mu_{r_5}$ are both disadvantaged on $W$-side, but $\mu_y$ corresponds to antichain $\{r_4, r_5\}$ deriving a $M$-disadvantaged stable marriage.

It follows that, for the former antichains, our objective is particularized as minimizing $C_W$, while for the latter antichains, it turns to minimize $C_M$. Next, we characterize the rotations contained in these antichains to delimit the search space i.e., we prune the pink hatched area in Figure 2 and extract $\mathcal{A}_I$ and $\mathcal{A}_{II}$, as Table 4

**Definition 2** (Layer rotations, $R[\ell]$)**.** *Let the layer of $r$, denoted as $L(r)$, be the length of longest path from $\mu_W$ to rotation $r$ on $G$. The set of layer-$\ell$ rotations is $R[\ell] = \{r \mid L(r) = \ell\}$.*

In particular, we can find the largest layer $\ell_W = \arg\max_{\substack{a = R[\ell], \\ \text{Worse}(\mu_a) = W\text{-side}}} \{\ell\}$, such that the antichain formed with layer-$\ell_W$ rotations derives a stable marriage disadvantaged on $W$-side.

Nonetheless, $G$ is not neatly divided in two subgraphs by Worse($\mu$) (Equation 3). We exploit $\mathcal{A}_I$ and $\mathcal{A}_{II}$ with candidate rotations by $\ell_W$.

**Definition 3** (Candidate rotations, $R_\triangleleft$)**.** *We define that antichain as $a_{\ell_W} = R[\ell_W]$ and the set of candidate rotations after layer $\ell_W$ that are still in $R_W$ as $R_\triangleleft = \{r \mid L(r) > \ell_W \wedge r \in R_W\}$.*

For example, in Figure 3 and Table 5, Worse($\mu_a$) = $W$-side with $a = \{r_1, r_2, r_3\}$ and Worse($\mu_a$) = $M$-side with $a = \{r_4, r_5, r_6\}$. As a result, $\ell_W = 1$ and $a_{\ell_W} = \{r_1, r_2, r_3\}$. $\mu_{r_4}, \mu_{r_5}$ and $\mu_{r_6}$ all have Worse($\mu_r$) = $W$-side. By Definition 3, we have $R_\triangleleft = \{r_4, r_5, r_6\}$.

We delimit the antichain sets $\mathcal{A}_I$ and $\mathcal{A}_{II}$ to antichains $a$ that contain at least one rotation $r \in R_\triangleleft$. By the following theorem, it suffices to search in $\mathcal{A}_I \cup \mathcal{A}_{II} \cup \{a_{\ell_W}\}$, hence we can eschew searching in the rest of $\mathcal{A}_W$, i.e., $\mathcal{A}_W \setminus (\mathcal{A}_I \cup \mathcal{A}_{II} \cup \{a_{\ell_W}\})$ (i.e., the pink hatched area in Figure 2).

**Theorem 2** (Sufficiency of $\mathcal{A}_I \cup \mathcal{A}_{II} \cup \{a_{\ell_W}\}$)**.** *An antichain in $\mathcal{A}_W \setminus (\mathcal{A}_I \cup \mathcal{A}_{II} \cup \{a_{\ell_W}\})$ cannot yield a better result in terms of balance cost than the optimal result found in $\mathcal{A}_I \cup \mathcal{A}_{II} \cup \{a_{\ell_W}\}$.*

*Proof.* By definition, any antichain $a' \in \mathcal{A}_W \setminus (\mathcal{A}_I \cup \mathcal{A}_{II} \cup \{a_{\ell_W}\})$ avoids rotations in layer $\ell_W$ and beyond, hence $\mu_{a'} \prec \mu_{a_{\ell_W}}$. By the definition of $\ell_W$ and $R_\triangleleft$, Worse($\mu_{a_{\ell_W}}$) = $W$-side. By Property 3.1, it is $C_W(\mu_{a_{\ell_W}}) < C_W(\mu_{a'})$, hence $\mu_{a_{\ell_W}}$ has better balance cost than $\mu_{a'}$. $\square$

---

[4]These antichains correspond to stable marriages resulting from rotation elimination $\mu_r / R_* \to \mu_a$, where $r \in R_W$ and $R_* \subset R_W$; by Property 1, starting with $C_M(\mu_r) < C_W(\mu_r)$, eliminating the rotations in $R_*$ increases $C_M(\mu_r)$ to $C_M(\mu_a)$ and decreases $C_W(\mu_r)$ to $C_W(\mu_a)$, where it may be $C_M(\mu_a) > C_W(\mu_a)$.

By Theorem 2, it suffices to search in $\mathcal{A}_\text{I}$ and $\mathcal{A}_\text{II}$ to generate any stable marriage better than $\mu_{a_{\ell_W}}$.

$\underline{\mathcal{A}_\text{III}}$ Next, we consider antichains $a$ that only contain rotations in $R_M$, $a \in \mathcal{A}_M$ in Figure 2 (blue area). Given such an antichain $a$, its corresponding stable marriage $\mu_a$ is either identical to, or may be derived from, an $\mu_r$ with[5] $r \in R_M$ by rotation elimination, $\mu_r/R_* \to \mu_a$. By Property 2, since $\mathsf{Worse}(\mu_r) = M$-side and $\mu_r \prec \mu_a$, it follows that $\mathsf{Worse}(\mu_a) = M$-side. Thus, we need only find the minimum $C_M$ in $\mathcal{A}_M$. We also delimit the rotations contained in these antichains to delimit $\mathcal{A}_\text{III}$, i.e., prune the blue hatched area in Figure 2 and extract $\mathcal{A}_\text{III}$ as Table 4.

**Definition 4** (Candidate rotations $R_\rhd$). *We define the set of candidate rotations in $R_M$ with all parents in $R_W$ as $R_\rhd = \{r|r \in R_M \wedge Parents(r) \subseteq R_W\}$.*

For example, in Figure 3, we first focus on rotations in $R_M$ and then extract $R_\rhd = \{r_7, r_8\}$. $R_\rhd$ does not contain $r_9$ and $r_{10}$, since both have a parent in $R_M$.

We delimit the antichain set $\mathcal{A}_\text{III}$ to antichains $a$ that contain only a single rotation, which is in $R_\rhd$, i.e., $a = \{r|r \in R_\rhd\}, |a| = 1$.

The following Theorem shows that it suffices to search in $\mathcal{A}_\text{III}$ as defined.

**Theorem 3** (Sufficiency of $\mathcal{A}_\text{III}$). *An antichain in $\mathcal{A}_M \setminus \mathcal{A}_\text{III}$ cannot yield a better result in terms of balance cost than the optimal result found in $\mathcal{A}_\text{III}$.*

*Proof.* Any antichain $a \in \mathcal{A}_M \setminus \mathcal{A}_\text{III}$ contains a rotation with a parent in $R_M$ or more than one rotation in $R_M$. Thus, we can generate $\mu_a$ by $\mu_r/R_* \to \mu_a$ with $r \in R_\rhd$. Since $\mathsf{Worse}(\mu_r) = M$-side and $\mu_r \prec \mu_a$, Property 3.2 implies that $C_M(\mu_r) < C_M(\mu_a)$. Thus, we can derive a more well-balanced stable marriage from antichains in $\mathcal{A}_\text{III}$. $\square$

**Other antichains.** The remaining antichains, $\mathcal{A}_{MW}$, contain rotations in both $R_M$ and $R_W$, i.e., the gray area in Figure 2. By virtue of Theorem 4, we ignore $\mathcal{A}_{MW}$ in the search process.

**Theorem 4** ($\mathcal{A}_\text{III}$ dominates $\mathcal{A}_{MW}$). *The antichain set $\mathcal{A}_{MW}$ cannot yield a stable marriage of better balance cost than the best stable marriage derived from $\mathcal{A}_\text{III}$.*

*Proof.* Any stable marriage $\mu$ corresponding to an antichain $a \in \mathcal{A}_{MW}$ derives as $\mu_r/R_* \to \mu_a$ with $r \in R_\rhd$. By Property 3.2, $C_M(\mu_a) > C_M(\mu_r)$, while $\mathsf{Worse}(\mu_r) = M$-side and $\mathsf{Worse}(\mu_a) = M$-side. Thus, an antichain in $\mathcal{A}_\text{III}$ yields a stable marriage more balanced than $\mu$. $\square$

In effect, $\mathcal{A}_\text{I}$, $\mathcal{A}_\text{II}$ and $\mathcal{A}_\text{III}$ suffice to find the exact solution to BSM, summarized in Table 3.

## 4.4 LOCAL SEARCH ALGORITHM

**Naïve approach (ENUM).** A naïve way to find the balanced stable marriage $\mu^*$ is to enumerate all stable marriages and apply Equation 2. An efficient algorithm for enumerating stable marriages, ENUM (Gusfield & Irving, 1989), expands a closed subset with a rotation in each step; its time complexity is $O(n^2 + nN)$. The pseudocode is detailed in Appendix A.2. As $N$ can be extremely large in some instances (Irving & Leather, 1986), we design an efficient and practical algorithm that reduces the search space.

---

**Algorithm 1** ISORROPIA

**Input:** Rotation Graph $G$
**Output:** Balanced Stable Marriage $\mu^*$
1: **if** $C_M(\mu_W) \geq C_W(\mu_W)$ **then return** $\mu_W$      ※ *check $\mu_W$*
2: **if** $C_W(\mu_M) > C_M(\mu_M)$ **then return** $\mu_M$      ※ *check $\mu_M$*
3: **for** $r \in R$ **do** Caculate $C_M(\mu_r)$ and $C_W(\mu_r)$
4: Collect candidate rotation subsets $R_\lhd$, $R_\rhd$      ※ *Definitions 3 and 4*
5: $\mu_\rhd^* \leftarrow$ LOCAL SEARCH IN $R_\rhd$      ※ *find the minimum $C_M$ in $\mathcal{A}_\text{III}$*
6: $\mu_\lhd^* \leftarrow$ LOCAL SEARCH IN $R_\lhd$      ※ *find the minimum $C_W$ in $\mathcal{A}_\text{I}$ and the minimum $C_M$ in $\mathcal{A}_\text{II}$*
7: **return** $\mu^* \leftarrow \mu_\lhd^*$ or $\mu_\rhd^*$ by Equation 2

---

**Our algorithm (ISORROPIA).** By the analysis in Section 4.3, ISORROPIA gathers two sets of candidate rotations $R_\lhd$ and $R_\rhd$ and searches three sets of promising antichains (Table 4) corresponding

---

[5]Note that $R_* \subseteq R_M$ is not necessary, since $r' \in a \setminus \{r\}$ may have predecessors in $R_W$ that are needed in $R_*$.

to a subset of all stable marriages to find the exact solution $\mu^*$. Algorithm 1 shows the pseudocode of ISORROPIA. First, Lines 1–2 check $\mu_M$ and $\mu_W$ for extreme cases. If $C_M(\mu_W) \geq C_W(\mu_W)$, we directly return $\mu_W$ as the optimal cost $\mu^*$, since, by Property 2, $C_M(\mu_W) < C_M(\mu)$ for any other stable matching $\mu$ (i.e., $\mu_W$ is pessimal for $W$-side and optimal for $M$-side). Symmetrically, if $C_W(\mu_M) > C_M(\mu_M)$, we return $\mu_M$. Otherwise, in the general case, we calculate $C_M(\mu_r)$ and $C_W(\mu_r)$ for all rotations, and collect the subsets $R_\triangleleft$ and $R_\triangleright$ (Lines 3–4) and find the locally optimal stable marriages $\mu_\triangleright^*$ in $\mathcal{A}_{\text{III}}$ and $\mu_\triangleleft^*$ in $\mathcal{A}_{\text{I}} \cup \mathcal{A}_{\text{II}} \cup \{a_{\ell_W}\}$ via local search on $R_\triangleright$ and $R_\triangleleft$. The pseudocode of local search can be found in Appendix A.3.

Overall, the time complexity is $O\left((|R| + \ell_W) \cdot n^2 + nN_\triangleleft\right)$, where $N_\triangleleft$ is the number of stable marriages enumerated in local search. While potentially exponential, as the problem is NP-hard, ISORROPIA reduces the search space $N$ to $N_\triangleleft + |R| + \ell_W$. The details of time cost can be found in Appendix A.3.

We extend ISORROPIA to find the *exact* sex-equal stable marriage, which calls to minimize the difference of satisfaction among two sides, detailed in Appendix A.4.

## 5 EXPERIMENTS

We compare the runtime and balance cost of ISORROPIA to those of baselines: (1) ENUM$^-$, our revised version of ENUM (Gusfield & Irving, 1989) (Section 4.4 and Appendix A.2), an enumeration algorithm on the rotation graph that returns the exact solution; (2) BILS (Viet et al., 2016b;a), a greedy local search method on the stable marriage lattice; we set the probability of random movement to $p = 0.05$; (3) DACC (Dworczak, 2016; 2021), Deferred Acceptance with Compensation Chains, a heuristic that finds a fair stable marriage by allowing proposals from both sides and ensuring the compensation of abandoned partners; (4)POWERBALANCE (Tziavelis et al., 2019), a heuristic that goes through a series of proposal iterations from both sides by *strongly deferred acceptance*, whereby unmatched agents only accept proposals more preferable than their own target, with the maximum number of proposal rounds fixed to $t = \left\lceil n \log_2^2 n / 10 \right\rceil$; (5) HMS (Tziavelis et al., 2019), a heuristic that improves upon the results of POWERBALANCE by an $m$-step local search over $k$ rounds, with complexity $O(tn + kmn^2)$. We emphasize that ENUM$^-$ and ISORROPIA (our algorithms) find the exact solution to BSM, while BILS, DACC, POWERBALANCE and HMS are only heuristics.

We use synthetic and real datasets in our assessment, as follows: (i) Following a prior work (Tziavelis et al., 2020), we construct a dataset, Uniform, with preference lists drawn from the uniform distribution. (ii) We use the settings in (Siala & O'Sullivan, 2017) to generate Hard instances by the method outlined in (Irving & Leather, 1986), which yields feasible stable marriages growing exponentially with $n$, hence instances of this family become unnaturally hard as $n$ grows; to ameliorate this hardness, we randomly pick 10% of individuals in each preference list and reshuffle their positions. (iii) Taxi reflects the two-sided market of taxis and users, drawn from the NYC Taxi dataset[6]; we define preferences for the two sides using distances and amounts. (iv) Adm captures a two-sided market of university admissions; we employ university rankings[7] and GRE and TOEFL scores to define[8] preferences on two sides. Table 6 presents the parameters and statistics of these datasets.

Table 6: Data sets: size $n$, rotations $|R|$ and edges $|E|$ in the rotation graph.

| | Uniform | | | Hard | | | Taxi | | | Adm | |
|---|---|---|---|---|---|---|---|---|---|---|---|
| $n$ | $|R|$ | $|E|$ | $n$ | $|R|$ | $|E|$ | $n$ | $|R|$ | $|E|$ | $n$ | $|R|$ | $|E|$ |
| 2.5k | 321 | 12,107 | 128 | 100 | 478 | 2k | 181 | 1,646 | 1k | 107 | 451 |
| 5k | 516 | 34,103 | 256 | 140 | 1,597 | 3k | 294 | 4,354 | 1.5k | 168 | 1,296 |
| 7.5k | 673 | 60,235 | 512 | 229 | 4,780 | 4k | 452 | 9,945 | 2k | 229 | 2,444 |
| 10k | 820 | 92,236 | 1024 | 362 | 13,834 | 5k | 579 | 15,761 | 2.5k | 318 | 4,806 |

We ran experiments on an Intel i5-13500H machine @2.60 GHz with 32G memory running Windows. All methods were implemented in C++; the code is available in our Github repository.[9]

Figure 4 presents our results on runtime and balance cost. As the brute-force method, ENUM$^-$, enumerates all feasible stable matchings $N$, to ensure termination we let it enumerate at most $10^7$

---

[6]https://www.nyc.gov/site/tlc/about/data.page
[7]https://kaggle.com/datasets/mylesoneill/world-university-rankings
[8]https://kaggle.com/datasets/akshaydattatraykhare/data-for-admission-in-the-university
[9]https://github.com/Asuka54089/Isorropia

stable marriages per instance. All methods are terminated after a time limit of 600 seconds. As POWERBALANCE and HMS perform very similarly in both balance cost and time, we report results for POWERBALANCE only for the sake of readability.

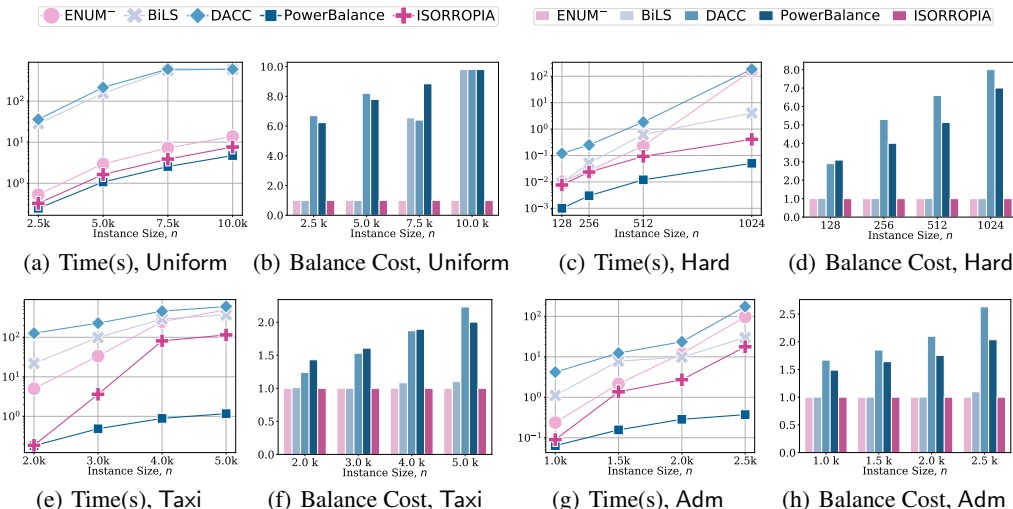

(a) Time(s), Uniform   (b) Balance Cost, Uniform   (c) Time(s), Hard   (d) Balance Cost, Hard

(e) Time(s), Taxi   (f) Balance Cost, Taxi   (g) Time(s), Adm   (h) Balance Cost, Adm

Figure 4: Time and balance cost (only ISORROPIA guarantees the exact solution.)

We observe the hardness of instances with different perferences. Instances in Uniform inherently have fewer feasible stable matchings than those in Hard, hence the brute-force enumeration on the rotation graph is time-consuming on hard instances. ISORROPIA decreases the search space by a factor of about $10^1$ to $10^4$ (detailed in Appendix A.5), improving runtime by a factor of 1 to ENUM$^-$(Figure 4(e)) and a factor of 2 to heuristics (Figure 4(a)). Further, ISORROPIA can be faster than DACC and BILS, but it is slower than heuristic algorithms. Even though instance sizes are only up to 1024 in these data sets, the number of stable matchings increases to $10^6$, yet ISORROPIA manages this increase in a scalable manner. For instances in the real spatial dataset, Taxi, the number of feasible stable matchings grows up to $10^6$, yet ISORROPIA improves runtime by a factor of up to $10^3$ (Figures 4(e)). For instances in Adm, the improvement is about a factor of $10^1$.

We compare the balance costs of stable matchings by ISORROPIA and baselines, by *the percentage of balance cost over the optimal*, in Figure 4. As we show in Section 4, ISORROPIA finds the exact solution to BSM. In the instances where ENUM$^-$ terminates naturally (i.e., $N \leq 10^7$) and returns the exact solution, ISORROPIA finds the stable marriage with same balance cost as ENUM$^-$. In instances where ENUM$^-$ terminates by the enumeration constraint (i.e., $N > 10^7$) and may not return the exact solution, ISORROPIA finds the best balance cost. The three heuristics, BILS, DACC and POWERBALANCE do not guarantee the balance cost, having a gap of at worst 9 times from the exact balance cost (Figure 4(b) and 4(h)). These heuristics may suffer heavy losses on the worst-off side satisfaction as $n$ increases, as the range of balance cost is $[n, n^2]$. For example, in Adm, $n = 2500$, the balance costs generated by ISORROPIA, BILS, DACC and POWERBALANCE are $273353, 300159(+26806), 718918(+445565)$, and $556764(+283411)$, respectively. ISORROPIA finds a stable marriage of optimal balance cost and also exhibits competitive time performance. To understand the internal workings of the search (discussed in Section 4.3), we report statistics on its operation in Appendix A.5.

# 6   CONCLUSION

We addressed the NP-hard problem of finding a fair stable matching that balances the satisfaction levels of both parties involved. As in real-world two-sided markets, the number of stable matchings can be large, an efficient traversal of the search space is imperative. We proposed an exact algorithm, ISORROPIA, that locally searches a reduced search space of three sets of antichains on the rotation graph. Our extensive experimental study demonstrates that ISORROPIA not only performs efficiently on synthetic and real datasets, including hard instances, but also, quite remarkably, outperforms in terms of time-efficiency heuristics that, as we also show, *do not* return an optimal balance cost.

ACKNOWLEDGMENTS

This work received support from the Science and Technology Development Fund Macau SAR (0003/2023/RIC, 0052/2023/RIA1, 0031/2022/A, 001/2024/SKL for SKL-IOTSC), the Research Grant of the University of Macau (MYRG2022-00252-FST), and the Shenzhen-Hong Kong-Macau Science and Technology Program Category C (SGDX202308821095159012). The work was also performed in part at SICC, which is supported by SKL-IOTSC, University of Macau. Additionally, this work was supported by an International Network Programme grant from the Danish Agency for Higher Education and Science.

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

# A   APPENDIX / SUPPLEMENTAL MATERIAL

## A.1   THE STRUCTURE OF ALL STABLE MARRIAGES

Given a stable marriage instance $\mathcal{I}$, all stable marriages $\mathcal{U}$ are composed of (1) two side-pessimal stable marriages ($\mu_W$ and $\mu_M$) and (2) other stable marriages. Figure 5 shows a conceptual framework of the structure of all stable marriages. First, two side-pessimal stable marriages can be generated by *Deferred Acceptance* (DA) algorithm upon its first termination. Then, other stable marriage can be generated by re-assigning some pairs from $\mu_W$, and finally it can reach at $\mu_M$. The re-assignment follows a set of DA procedures (i.e., break stable marriages and apply DA multiple times), which can be compactly represented by a set of rotation nodes.

Table 7: Notations

| Notation | Description |
|---|---|
| $\mu_M, \mu_W$ | $[M]$-pessimal and $[W]$-pessimal stable marriages |
| $C_M, C_W$ | dissatisfactions of side $[M]$ and $[W]$(Equation 1) |
| $R$ | rotation poset |
| $G$ | rotation graph |
| $Pred(r)$ | predecessors of $r$ |
| $a, s, \mu$ | an antichain, a closed subset and a stable marriage |
| $\mathcal{A}, \mathcal{S}, \mathcal{U}$ | sets of all antichains, closed subsets and stable marriages |
| $n$ | instance size, i.e., size of agent sets $M$ and $W$ |
| $N$ | number of all stable marriages, i.e., size of $\mathcal{U}$ |
| $\mu_a$ | a stable marriage derived from the antichain $a$ |
| $a_r, s_r, \mu_r$ | $r$-related antichain, closed subset and stable marriage |
| $R_M, R_W$ | the $[M]/[W]$-disadvantaged rotations (Definition 1) |
| $\mathcal{A}_M, \mathcal{A}_W, \mathcal{A}_{MW}$ | three subsets of $a$ divided by disadvantaged rotations |
| $\mathcal{A}_I, \mathcal{A}_{II}, \mathcal{A}_{III}$ | three sets of promising antichains (Table 4) |
| $D_{out}$ | out-degree list of all rotations |
| $R_0$ | double-ended queue of rotations in running closed subset |
| $\mu_\triangleleft^*, \mu_\triangleright^*$ | two local optimal stable marriages (Table 4) |

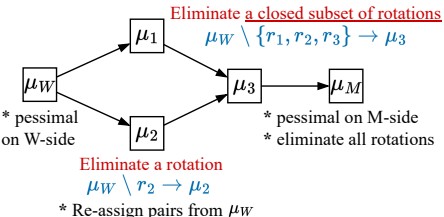

Figure 5: The structure of all stable marriages

**DA algorithm.** The *Deferred Acceptance* (DA) algorithm (Gale & Shapley, 1962) lets each man $m$ start from the first preference and sequentially propose to the next most preferable woman in the order of $P_m$, as long as the man finds itself being single. Each woman $w$ accepts a $(m, w)$ proposal if the woman is single or prefers $m$ to the current partner $\mu(w)$.

**Example 1** (All stable marriages). *Given the instance in Table 8, we discuss four DA procedures that generates all stable marriages in Table 10. The corresponding re-assigned pairs of each DA procedure are highlighted respectively with black, green, red and blue boxes in Table 8 and Table 10.*

- *First, we start the proposal sequence: (1) $m_1$ to $w_5$ (accepts $m_1$); (2) $m_2$ to $w_2$ (accepts $m_2$); (3) $m_3$ to $w_5$ (accepts $m_3$, abandons $m_1$); (4) $m_1$ to $w_1$ (accepts $m_1$), etc. When the DA algorithm first terminates (i.e., each man gets assigned), we get the stable marriage $\mu_0$ in Table 10 and the placements on preference lists are highlighted with the black box in Table 8.*

- *Next, if we break $\langle m_1, w_1 \rangle$ (or $\langle m_5, w_2 \rangle$) in $\mu_0$, and then apply DA algorithm that makes $m_1$ proposes to $w_2$ (or $m_5$ proposes to $w_2$), we re-assign pairs (highlighted with the green box) from $\mu_0$ and get a new stable matching $\mu_1$. If we break any pair of $\langle m_2, w_5 \rangle\langle m_3, w_3 \rangle\langle m_4, w_4 \rangle$ in $\mu_0$, and then apply DA algorithm, we re-assign pairs (highlighted with the red box) from $\mu_0$ and get the stable marriage $\mu_2$.*

- *We can re-assign pairs highlighted with red and green box together from $\mu_0$ and get the stable marriage $\mu_3$. Further, based on $\mu_3$, by breaking $\langle m_4, w_5 \rangle$ or $\langle m_5, w_1 \rangle$, and then applying DA algorithm, we can re-assign pairs (highlighted with the blue box) from $\mu_3$ and get the stable marriage $\mu_4$. We can no longer apply DA algorithm for $\mu_4$, since each woman has no better choices. In other words, woman are unwilling to accept new proposals.*

The DA algorithm (Gale & Shapley, 1962) outputs a stable marriage optimal for each agent on one side and pessimal for each agent on the other side (McVitie & Wilson, 1971; Irving & Leather, 1986), i.e., we get $\mu_0$ if men propose to women and we get $\mu_4$ if women propose to men. Shown in Table 10, we denote these two outputs as $\mu_W$ and $\mu_M$, where the subscript denotes the side that gets a pessimal outcome.

**Domination relationships.** A stable marriage $\mu$ *dominates* another stable marriage $\mu'$, or $\mu \prec \mu'$, if each agent on the $M$-side gets a no less preferable partner in $\mu$ than in $\mu'$, and, as stability implies, each agent on the $W$-side gets a no more preferable partner in $\mu$ than in $\mu'$. The set of all stable marriages forms a distributive lattice (Gusfield & Irving, 1989), where $\mu_W$ dominates, and $\mu_M$ is dominated by, any other stable marriage. Meanwhile, no stable marriage can achieve a better choice ahead of the black boxes for men and a worse choice afterwards the black boxes for women.

**Rotation elimination.** To compactly represent the breakable pairs and the corresponding re-assigned pairs for each DA process, we use the construct of *rotation* (Irving, 1985; Irving & Leather, 1986). A rotation

Table 8: Preference lists

| **Preference lists of men** | | | | | |
|---|---|---|---|---|---|
| $P_{m_1}$ | $w_5$ | $w_1$ | $w_2$ | $w_4$ | $w_3$ |
| $P_{m_2}$ | $w_2$ | $w_5$ | $w_3$ | $w_4$ | $w_1$ |
| $P_{m_3}$ | $w_5$ | $w_3$ | $w_4$ | $w_2$ | $w_1$ |
| $P_{m_4}$ | $w_4$ | $w_5$ | $w_3$ | $w_1$ | $w_2$ |
| $P_{m_5}$ | $w_4$ | $w_2$ | $w_1$ | $w_5$ | $w_3$ |

| **Preference lists of women** | | | | | |
|---|---|---|---|---|---|
| $P_{w_1}$ | $m_2$ | $m_4$ | $m_3$ | $m_5$ | $m_1$ |
| $P_{w_2}$ | $m_1$ | $m_5$ | $m_3$ | $m_4$ | $m_2$ |
| $P_{w_3}$ | $m_2$ | $m_4$ | $m_1$ | $m_5$ | $m_3$ |
| $P_{w_4}$ | $m_3$ | $m_1$ | $m_4$ | $m_5$ | $m_2$ |
| $P_{w_5}$ | $m_5$ | $m_4$ | $m_2$ | $m_3$ | $m_1$ |

Table 9: Rotations

| $r$ | After elimination | $* \to r$ |
|---|---|---|
| $r_1 = \langle m_1, w_1 \rangle \langle m_5, w_2 \rangle$ | $\langle m_1, w_2 \rangle \langle m_5, w_1 \rangle$ | |
| $r_2 = \langle m_2, w_5 \rangle \langle m_3, w_3 \rangle \langle m_4, w_4 \rangle$ | $\langle m_2, w_3 \rangle \langle m_3, w_4 \rangle \langle m_4, w_5 \rangle$ | |
| $r_3 = \langle m_4, w_5 \rangle \langle m_5, w_1 \rangle$ | $\langle m_4, w_1 \rangle \langle m_5, w_5 \rangle$ | $r_1, r_2$ |

Figure 6: $G$, Rotation Graph

Table 10: Rotation elimination and balance costs for all stable marriages

| $\mu$ | Matches | | | | $a$ | $s$ | $C_M(\mu)$ | $C_W(\mu)$ | Worse$(\mu)$ | Balance Cost |
|---|---|---|---|---|---|---|---|---|---|---|
| $\mu_0(\mu_W)$ | $\langle m_1,w_1 \rangle \langle m_2,w_5 \rangle \langle m_3,w_3 \rangle \langle m_4,w_4 \rangle \langle m_5,w_2 \rangle$ | | | | $\emptyset$ | $\emptyset$ | 9 | 18 | $W$-side | 18 |
| $\mu_1$ | $\langle m_1,w_2 \rangle$ | $\langle m_2,w_5 \rangle \langle m_3,w_3 \rangle \langle m_4,w_4 \rangle$ | $\langle m_5,w_1 \rangle$ | | $\{r_1\}$ | $\{r_1\}$ | 11 | 16 | $W$-side | 16 |
| $\mu_2$ | $\langle m_1,w_1 \rangle$ | $\langle m_2,w_3 \rangle \langle m_3,w_4 \rangle \langle m_4,w_5 \rangle$ | $\langle m_5,w_2 \rangle$ | | $\{r_2\}$ | $\{r_2\}$ | 12 | 11 | $M$-side | 12 |
| $\mu_3$ | $\langle m_1,w_2 \rangle$ | $\langle m_2,w_3 \rangle \langle m_3,w_4 \rangle \langle m_4,w_5 \rangle$ | $\langle m_5,w_1 \rangle$ | | $\{r_1,r_2\}$ | $\{r_1,r_2\}$ | 14 | 9 | $M$-side | 14 |
| $\mu_4(\mu_M)$ | $\langle m_1,w_2 \rangle$ | $\langle m_2,w_3 \rangle \langle m_3,w_4 \rangle$ | $\langle m_4,w_1 \rangle \langle m_5,w_5 \rangle$ | | $\{r_3\}$ | $\{r_1,r_2,r_3\}$ | 17 | 6 | $M$-side | 17 |

belonging to (or *exposed in*) $\mu$ is an ordered sub-list of matched pairs $r = \{\langle m_i, \mu(m_i) \rangle, \langle m_{i+1}, \mu(m_{i+1}) \rangle, \ldots, \langle m_{i+d}, \mu(m_{i+d}) \rangle\}$. Given a $\mu$ that exposes a rotation $r$, we can break the marriage of $m_i$ in rotation $r$ and apply the DA algorithm to let $m_i$ propose to the next most preferable user, eventually being assigned with user $\mu(m_{i+1})$ who abandons task $m_{i+1}$; likewise, $m_{i+1}$ will then be matched with $\mu(m_{i+2})$, and so on until we reach $\mu(m_i)$ in full cycle. Intuitively, each of the tasks $m_i, m_{i+1}, \ldots, m_{i+d}$ is matched to a user less preferable to it, $\mu(m_{i+1}), \mu(m_{i+2}), \ldots, \mu(m_i)$ respectively, while each of the user $\mu(m_i), \mu(m_{i+1}), \ldots, \mu(m_{i+d})$ is matched to a task more preferable to the user, $m_{i+d}, m_i, \ldots, m_{i+d-1}$ respectively. Thus, the ensuing matching $\mu'$ is still stable. We call this re-coupling *rotation elimination*, denoted as $\mu / r \to \mu'$. By *eliminating* the rotation $r$, we can obtain a new stable marriage $\mu'$. Certainly, $\mu$ dominates $\mu'$.

**Example 2** (Rotation Elimination). *Table 9 shows the breakable pairs and the corresponding re-assigned pairs of rotations $r_2$, exposed in the $W$-pessimal stable marriage $\mu_W$. It is computed by DA algorithm and follows the movements from black boxes to red boxes in Table 8. For $m_2, m_3, m_4$, each of them gets a less preferable choice (i.e., the movement of boxes are from high to low), while each of $w_5, w_3, w_4$ gets a more preferable choice (i.e., the movement of boxes are from low to high). As shown in Table 10, after eliminating $r_2$ from $\mu_W$, $\mu_W / r_2$, we get a new stable marriage $\mu_2$.*

**Eliminating a set of rotations.** The set of all rotations $R$ constitutes a *partially ordered set* (poset) $(R, \to)$ (Irving & Leather, 1986). A partial order $r \to r'$ indicates that $r'$ is only exposed *after* eliminating $r$. The poset $(R, \to)$ is represented by a directed acyclic graph $G = (R, E)$, where nodes stand for rotations and an edge $(r, r')$ denotes a *direct* partial order $r \to r'$, i.e., $\nexists r_* | r \to r_* \to r'$, guaranteeing connectivity. We denote the predecessors of $r$ as $Pred(r)$, i.e., for each $r' \in Pred(r)$, $r$ is reachable from $r'$.

Finding all rotations and all edges in $G$ cost $O(n^2)$ time (Irving et al., 1987; Gusfield, 1987); this construction is a preprocessing step beneath the core of our problem.

**Example 3** (Poset and Rotation Graph). *Figure 6 shows an example of rotation graph $G$ with partial order relationships representing a rotation poset. As rotations $r_1$ and $r_2$ are not predecessors of each other, we may eliminate them in an arbitrary order, as $\mu_W / \{r_1, r_2\}$ or $\mu_W / \{r_2, r_1\}$. However, as both $r_1$ and $r_2$ are predecessors of $r_3$, we expose $r_3$ only after we eliminate both $r_1$ and $r_2$, i.e., $r_3$ is exposed in stable marriage $\mu_3$ but not in $\mu_1$ and $\mu_2$, as shown in Table 10.*

**Antichains and Closed Subsets.** An *antichain* $a$ is a subset of $R$ such that no rotation in $a$ is a predecessor of another. Given an antichain $a$, we can construct a unique *closed subset* $s = \bigcup_{r \in a} \{r\} \cup Pred(r)$, which contains all rotations in $a$ and their predecessors. Recall that for $a = \{r_3\}$ in Example 3, we should eliminate

rotations in its closed subset, $s = \{r_1, r_2, r_3\}$, according to partial order relationship starting from $\mu_W$, so that each rotation is exposed in the elimination process.

**Example 4** (Antichain and Closed Subset). *As shown in Figure 6 and Table 10, $\{r_1, r_2\}$ is an antichain and $\{r_1, r_2\}$ is its corresponding closed subset. A single rotation also forms an antichain, e.g., $a = \{r_3\}$ corresponds to $s = \{r_1, r_2, r_3\}$. A counterexample of an antichain is $\{r_1, r_3\}$, where $r_1$ is a predecessor of $r_3$.*

**All Stable Marriages.** Let $\mathcal{A}$ be the set of antichains and $\mathcal{S}$ the set of closed subsets in $G$. Theorem 1 (Irving & Leather, 1986) provides a foundational fact on the structure of all stable marriages.

**Theorem 5** (Relationship between antichains, closed subsets, and stable marriages). *(Irving & Leather, 1986) In any stable marriage instance there is a one-to-one relationship among antichains $\mathcal{A}$, closed subsets $\mathcal{S}$ and stable marriages $\mathcal{U}$. Enumerating all stable marriages is #P-complete.*

In other words, for any antichain $a$, we can find a corresponding closed subset $s$ and stable marriage $\mu$ via rotation elimination $\mu_W / s \to \mu_a$. For simplicity, we refer to these concepts (i.e., $\mu$, $a$, and $s$) interchangeably without loss of clarity. As all stable marriages listed in Table 10, we can calculate the side dissatisfactions ($C_M$ and $C_W$), the worse-off side ($\mathsf{Worse}(\mu)$) and the balance cost by Equation 1, 3 and 2 respectively. The balanced stable marriage is $\mu_2$.

## A.2 NAÏVE APPROACH (ENUM AND ENUM$^-$)

A naïve way to find the balanced stable marriage $\mu^*$ is to enumerate all stable marriages and apply Equation equation 2. An efficient algorithm for enumerating stable marriages, ENUM (Gusfield & Irving, 1989), expands a closed subset with a rotation in each step; its time complexity is $O(n^2 + nN)$. As $N$ can be extremely large in some instances (Irving & Leather, 1986), we design an efficient and practical algorithm that reduces the search space.

Unfortunately, we cannot use ENUM to find promising antichains either, as it enumerates closed subsets in an order from $\emptyset$ to $R$ (i.e., from $\mu_W$ to $\mu_M$). Intuitively, the enumeration has a tendency from dominating stable marriages to dominated stable marriages. By Property 3.1, unpromising antichains will be enumerated before promising antichains. To overcome this problem, we reverse the enumeration order of ENUM to craft ENUM$^-$, which enumerates closed subsets from $R$ to $\emptyset$ (i.e., from $\mu_M$ to $\mu_W$), thus benefits from pruning $\mathcal{A}_M$ to $\mathcal{A}_{\mathrm{III}}$ (i.e., by Table 4, $\mathcal{A}_{\mathrm{III}}$ only consists the antichains of length 1), and devise our approach based on ENUM$^-$. In this process, we say that a rotation node $r$ having out-degree 0 in a subgraph is a *terminal* node therein; the set of terminal nodes within a closed subset $s$ is the antichain corresponding to $s$.

---

**Algorithm 2** ENUM$^-$

**Input:** Rotation Graph $G$
**Output:** Balanced Stable Marriage $\mu^*$
1: Initialize $D_{out}$, $R_0$, $s := R$
2: ENUMERATE$(s, D_{out}, R_0)$       ※ *start enumeration from G*
3: **return** $\mu^*$       ※ *return the exact BSM solution*
4: **function** ENUMERATE$(s, D_{out}, R_0)$
5:      **if** $R_0 \neq \emptyset$ **then**
6:          $r := R_0.\text{pop\_front}$
7:          $s.\text{remove}(r)$, $s \xrightarrow[\text{elimination}]{\text{rotation}} \mu$       ※ *a new closed subset $s \setminus \{r\}$*
8:          Update $\mu^*$ to $\mu$ by Equation 2
9:          **for** $r' \in parents(r)$ **do**
10:             $D_{out}(r') := D_{out}(r') - 1$       ※ *out-neighbor of $r'$ removed*
11:             **if** $D_{out}(r') = 0$ **then** $R_0.\text{push\_back}(r')$
12:          ENUMERATE$(s, D_{out}, R_0)$       ※ *recursive call (i)*
13:          **for** $r' \in parents(r)$ **do**
14:             $D_{out}(r') := D_{out}(r') + 1$
15:             **if** $D_{out}(r') = 1$ **then** $R_0.\text{pop\_back}$
16:          $s.\text{add}(r)$
17:          ENUMERATE$(s, D_{out}, R_0)$       ※ *recursive call (ii)*
18:          $R_0.\text{push\_front}(r)$

---

Algorithm 2 shows the pseudocode of ENUM$^-$. Using an array $D_{out}$ to record the out-degree of all rotations in a shrinking rotation graph and a double-ended queue $R_0$ to store the running terminal nodes in closed subset $s$, ENUM$^-$ recursively performs two operations: (i) it removes a terminal node $r \in R_0$ from the running closed subset $s$ (Line 7), reduces the out-degrees $D_{out}$ of parent nodes and enters them to $R_0$ if they become terminal

nodes thereby (Lines 9–11), and proceeds recursively (Line 12); and (ii) it restores the out-degrees $D_{out}$ and any corresponding terminal nodes from $R_0$ to $s$ (Lines 13–15) and the previously removed $r$ to $s$ (Line 16), proceeds to recursively remove from $s$ other terminal nodes in $R_0$ (Line 17), and eventually restores $r$ to $R_0$ (Line 18).

## A.3 LOCAL SEARCH IN ISORROPIA

The pseudocode of local search strategies of ISORROPIA is shown in Algorithm 3. In particular:

- Local Search in $R_\rhd$. As Theorem 3 shows and Table 4 illustrates, to find the locally optimal stable marriage $\mu_\rhd^*$ in $\mathcal{A}_{\text{III}}$ that minimizes $C_M$, we only need to consider all $\mu_r$ with $r \in R_\rhd$.

- Local Search on $R_\lhd$. First, we set $\mu_\lhd^*$ to $\mu_{a_{\ell_W}}$ and update the balance cost by Equation equation 4 (Line 4). Then we create the subgraph of $G$ induced by $R_W$ (Lines 5–6), which corresponds to $\mathcal{A}_W$, and starting with $s = R_W$, enumerate closed subsets $s$, hence stable marriages $\mu$, while keeping track of their corresponding antichains $a$. This enumeration proceeds while the antichain $a$ contains at least one rotation in $R_\rhd$ (Line 10), hence belongs to $\mathcal{A}_{\text{I}} \cup \mathcal{A}_{\text{II}}$. The enumeration removes rotations from $s$, progressively generating a stable marriage $\mu'$ from another $s$, such that $\mu' \prec \mu$. Thus, if $\text{Worse}(\mu) = W$-side, by Properties 3.1, it is $C_W(\mu) < C_W(\mu')$ and $\mu'$ is better. In effect, it terminates when it reaches a stable marriage $\mu_a$ with $\text{Worse}(\mu_a) = W$-side while the best solution at hand is better than $\mu_a$ (Line 11).

---

**Algorithm 3** Local Search Strategies

1: **function** LOCAL SEARCH IN $R_\rhd$
2:      **return** $\min\limits_{r \in R_\rhd} C_M(\mu_r)$          ✳ *$|a| = 1$ for $a \in \mathcal{A}_{\text{III}}$*
3: **function** LOCAL SEARCH IN $R_\lhd$
4:      Update $\mu_\lhd^*$ using $\mu_{a_{\ell_W}}$
5:      Initialize $D_{out}, R_0$ for $R_W$          ✳ *the subgraph of $G$ corresponding to $\mathcal{A}_W$*
6:      ENUMERATE($R_W, R_0, D_{out}, R_0$)          ✳ *$\mathcal{A}_{\text{I}} \cup \mathcal{A}_{\text{II}}$*
7:      **return** $\mu_\lhd^*$
8: **function** ENUMERATE($s, a, D_{out}, R_0$)
9:      **if** $R_0 \neq \emptyset$ **then**
10:          **if** $a \cap R_\lhd = \emptyset$ **then return**          ✳ *antichain must contain $r \in R_\lhd$*
11:          **if** $\text{Worse}(\mu_a) = W$-side and $C_W(\mu_a) \geq C(\mu_\lhd^*)$ **then return**
12:          $r := R_0.\text{pop\_front}$
13:          $s.\text{remove}(r), a.\text{remove}(r), s \to \mu$          ✳ *new closed subset $s \setminus \{r\}$*
14:          Update $\mu_\lhd^*$ using $\mu$
15:          **for** $r' \in parents(r)$ **do**
16:              $D_{out}(r') := D_{out}(r') - 1$
17:              **if** $D_{out}(r') = 0$ **then** $R_0.\text{push\_back}(r'), a.\text{add}(r')$
18:          ENUMERATE($s, a, D_{out}, R_0$)          ✳ *recursive call (i)*
19:          **for** $r' \in parents(r)$ **do**
20:              $D_{out}(r') := D_{out}(r') + 1$
21:              **if** $D_{out}(r') = 1$ **then** $R_0.\text{pop\_back}, a.\text{remove}(r')$
22:          $s.\text{add}(r), a.\text{add}(r), s \to \mu$
23:          ENUMERATE($s, a, D_{out}, R_0$)          ✳ *recursive call (ii)*
24:          $R_0.\text{push\_front}(r)$

---

**Time Cost.** The time cost of ISORROPIA is dominated by (i) gathering candidate rotations subsets $R_\lhd$ and $R_\rhd$ and (ii) searching in those. First, we gather $R_M$ and $R_W$ by calculating $\mu_r$ for all rotations in $O(|R| \cdot n^2)$. We avoid calculating all $r$-related stable marriages, since a rotation is in $R_M$ if its parent is in $R_M$ (Property 2). In practice, we calculate about 50% of $\mu_r$ constructs, as detailed in Section 5. In $R_\lhd$, we find the maximal layer $\ell_W$ in $O(\ell_W \cdot n^2)$ deriving $a = R[1], R[2], \ldots, R[\ell_W], R[\ell_W + 1]$. Since there are no more layers than rotations, it is $\ell_W \leq |R|$. In $R_\lhd$, we check at most $n/2$ parents of each rotation in $R_M$ in $O(|R_M| \cdot n)$. Thus, this step requires $O\left((|R| + \ell_W) \cdot n^2\right)$ in total, where $|R| + \ell_W \ll N$ in most cases. Then, the local search in $R_\rhd$ only scans $r$-related stable marriages $\mu_r$ in $R_\rhd$, already calculated in the previous step (to collect $R_M$ and $R_W$), in $O(|R_\rhd|)$. By definition, $|R_\rhd|$ cannot be larger than $n/2$, the width of $G$. On the other hand, for local search in $R_\lhd$ we apply ENUM$^-$ with time complexity $O(n^2 + nN_\lhd)$, where $N_\lhd$ is the number of stable marriages enumerated in that local search. Overall, the time complexity is $O\left((|R| + \ell_W) \cdot n^2 + nN_\lhd\right)$. While potentially exponential, as the problem is NP-hard, ISORROPIA reduces the search space $N$ to $N_\lhd + |R| + \ell_W$, with $N_\lhd < 50\%N$ in practice.

## A.4 APPLICATION TO SESM

Here we apply ISORROPIA to the sex-equal stable marriage problem (SESM) (Kato, 1993). As Table 1 shows, the SESM and BSM problems have different objectives defined in terms of $C_M$ and $C_W$.

$$C_{se}(\mu) = \min_{\mu \in \mathcal{U}} |C_M(\mu) - C_W(\mu)| \tag{5}$$

We rewrite the sex-equality cost using the notation of Equation 4 as:

$$\mu_{se}^* = \arg\min_{\mu \in \mathcal{U}} \begin{cases} C_W(\mu) - C_M(\mu) & \text{if } \mathsf{Worse}(\mu) = W\text{-side} \\ C_M(\mu) - C_W(\mu) & \text{if } \mathsf{Worse}(\mu) = M\text{-side} \end{cases} \tag{6}$$

This minimization problem is also non-convex and NP-hard (Tziavelis et al., 2019). Yet we can apply ISORROPIA to SESM using the objective function in Equation 6, to find the exact solution for SESM. The following result ensures the correctness of ISORROPIA for SESM, namely that $\mathcal{A}_\mathrm{I}$, $\mathcal{A}_\mathrm{II}$ and $\mathcal{A}_\mathrm{III}$ remain promising antichains.

**Theorem 6** ($\mathcal{A}_\mathrm{I}$, $\mathcal{A}_\mathrm{II}$ and $\mathcal{A}_\mathrm{III}$ for SESM). *Theorems 2, 3, and 4 apply to SESM.*

*Proof.* In the proof of Theorem 2, we have $\mu_{a'} \prec \mu_{a_{\ell_W}}$. By Property 1 and Property 2, it is $\mathsf{Worse}(\mu_{a'}) = W$-side, $\mathsf{Worse}(\mu_{a_{\ell_W}}) = W$-side, $C_W(\mu_{a'}) > C_W(\mu_{a_{\ell_W}})$, and $C_M(\mu_{a'}) < C_M(\mu_{a_{\ell_W}})$. Therefore, $C_W(\mu_{a'}) - C_M(\mu_{a'}) > C_W(\mu_{a_{\ell_W}}) - C_M(\mu_{a_{\ell_W}})$, hence $\mu_{a_{\ell_W}}$ has better sex-equality cost than $\mu_{a'}$. In the proof of Theorems 3 and 4, we infer that $\mu_r \prec \mu_a$, $\mathsf{Worse}(\mu_r) = M$-side, $\mathsf{Worse}(\mu_a) = M$-side, and $C_M(M_r) < C_M(\mu_a)$. Thanks to Property 1, it is $C_W(\mu_r) > C_W(\mu_a)$, hence $C_M(\mu_a) - C_W(\mu_a) > C_M(\mu_r) - C_W(\mu_r)$. □

## A.5 STATISTICS ON ISORROPIA

To understand the internal working of the search (detailed in Section 4.3), we also report statistics on the operation of ISORROPIA. Table 11 reports statistics on rotations and stable marriages, as the following average percentages scores per instance size:

- $|\mathcal{U}_{(M)}|/N, |\mathcal{U}_{(W)}|/N$: percentage of stable matchings having $\mathsf{Worse}(\mu) = M$-side and $\mathsf{Worse}(\mu) = W$-side as the worst case.
- $\#\mu/N$: $\#\mu$ is the number of stable matchings explored by ISORROPIA. Clearly, $\#\mu > N_\triangleleft$.
- $|\mathcal{A}_\mathrm{I}|/|\mu_{(W)}|, |\mathcal{A}_\mathrm{II}|/|\mu_{(M)}|$ and $|\mathcal{A}_\mathrm{III}|/|\mu_{(M)}|$: percentage of $\mathcal{A}_\mathrm{I}$ among $\mu_{(W)}$, $\mathcal{A}_\mathrm{II}$ among $\mu_{(M)}$, and $\mathcal{A}_\mathrm{III}$ among $\mu_{(M)}$. (Table 4)
- $|R_M|/|R|, |R_W|/|R|$: percentage of the two sets of disadvantaged rotations among all rotations. (Definition 1)
- $|R_\triangleleft|/|R_W|, |R_\triangleright|/|R_M|$: percentage of candidate rotations $R_\triangleleft$ among $R_W$ and $R_\triangleright$ among $R_M$. (Definition 3 and 4)

Table 11: Statistics of rotations and stable marriages

| Dataset | n | $|R|$ | $|R_W|/|R|$ | $|R_M|/|R|$ | $|R_\triangleleft|/|R_W|$ | $|R_\triangleright|/|R_M|$ | N | $|\mathcal{U}_{(M)}|/N$ | $|\mathcal{U}_{(W)}|/N$ | $\#\mu/N$ | $|\mathcal{A}_\mathrm{I}|/|\mu_{(W)}|$ | $|\mathcal{A}_\mathrm{II}|/|\mu_{(M)}|$ | $|\mathcal{A}_\mathrm{III}|/|\mu_{(M)}|$ |
|---|---|---|---|---|---|---|---|---|---|---|---|---|---|
| | 2.5k | 322 | 49.92% | 50.08% | 0.52% | 0.31% | 3,436 | 55.74% | 44.26% | 3.25% | 5.54% | 0.12% | 0.06% |
| Uniform | 5k | 516 | 50.09% | 49.91% | 0.44% | 0.21% | 7,243 | 50.97% | 49.03% | 4.07% | 9.47% | 0.19% | 0.04% |
| | 7.5k | 673 | 50.25% | 49.75% | 0.36% | 0.15% | 10,534 | 45.04% | 54.96% | 3.21% | 5.60% | 0.19% | 0.02% |
| | 10k | 821 | 49.83% | 50.17% | 0.33% | 0.17% | 13,420 | 48.21% | 51.79% | 3.76% | 6.77% | 0.06% | 0.02% |
| | 128 | 100 | 50.51% | 49.49% | 0.65% | 1.30% | 648 | 49.18% | 50.82% | 3.66% | 6.06% | 0.00% | 0.42% |
| Hard | 256 | 140 | 49.32% | 50.68% | 1.01% | 0.74% | 1,241 | 50.99% | 49.01% | 0.88% | 1.39% | 0.08% | 0.26% |
| | 512 | 229 | 48.78% | 51.22% | 1.68% | 0.44% | 13,227 | 40.58% | 59.42% | 0.30% | 0.55% | 0.15% | 0.03% |
| | 1024 | 362 | 50.78% | 49.22% | 0.96% | 0.36% | 3,426,000 | 35.40% | 64.60% | 0.18% | 0.20% | 5.00E-06 | 0.02% |
| | 2k | 181 | 55.05% | 44.95% | 2.82% | 0.75% | 74,804 | 51.08% | 48.92% | 19.79% | 37.91% | 0.69% | 0.05% |
| Taxi | 3k | 294 | 46.35% | 53.65% | 10.74% | 0.40% | 445,819 | 49.01% | 50.99% | 14.66% | 30.16% | 1.52% | 0.01% |
| | 4k | 453 | 52.39% | 47.61% | 2.37% | 0.27% | 1,572,916 | 45.27% | 54.73% | 38.58% | 47.76% | 2.73% | 3.14E-05 |
| | 5k | 580 | 52.78% | 47.22% | 7.15% | 0.29% | 1,977,611 | 36.81% | 63.19% | 48.91% | 66.13% | 3.37% | 9.79E-06 |
| | 1k | 108 | 54.03% | 45.97% | 4.38% | 1.08% | 17,826 | 40.68% | 59.32% | 18.39% | 31.84% | 0.47% | 0.12% |
| Adm | 1.5k | 169 | 50.96% | 49.04% | 4.23% | 0.70% | 97,949 | 50.20% | 49.80% | 21.64% | 34.60% | 6.83% | 0.02% |
| | 2k | 230 | 51.94% | 48.06% | 7.11% | 0.52% | 339,197 | 32.77% | 67.23% | 14.87% | 22.34% | 0.82% | 8.33E-05 |
| | 2.5k | 319 | 53.76% | 46.24% | 2.39% | 0.35% | 1,256,356 | 47.25% | 52.75% | 24.38% | 40.56% | 3.42% | 2.25E-05 |

In accordance with the analysis of time cost in Section 4.4, the number of rotations is smaller than the number of all stable marriages, i.e., $|R| + \ell_W \ll N$. We note that the two sets of disadvantaged rotations are almost equal-sized. ISORROPIA filters out most rotations to extract the candidate rotations $R_\triangleleft$ and $R_\triangleright$, which are about 0.1% to 10% of all rotations.

On the other hand, among stable matchings, the two worst cases are not so evenly shared as the two sets of rotations in Hard and Adm. Further, the sets of promising antichains comprise only a small subset of all stable matchings (Table 11, columns 12, 13, 14). In effect, ISORROPIA drastically reduces the search space and thereby improves upon efficiency. In effect, based on the compact representation of a rotation graph, we extract hidden relationships among stable matchings than would have been require to explore the stable matching lattice (Irving et al., 1987). Notably, we stop the enumeration of stable matchings in ENUM and ENUM$^-$ when their number exceeds $10^7$, hence do not report the results of these instances in Table 11.

