# OpenReview forum: "Balancing Bias in Two-sided Markets for Fair Stable Matchings"
_ICLR.cc/2025/Conference — ICLR 2025 Poster_

### Official Review · Reviewer_fH5T · 2024-10-26

**Soundness:** 4
**Presentation:** 4
**Contribution:** 3
**Rating:** 6
**Confidence:** 4

**Summary:**

This paper considers the Balanced Stable Marriage (BSM) problem. In a two-sided market where each agent from one side has a preference list over agents from the other side, a stable matching is a matching in which no pair of agents prefers each other over their current partners. The seminal Gale-Shapley algorithm finds a stable matching in polynomial time. Subsequent work has focused on finding a stable matching that is optimal in some measure, such as regret cost, egalitarian cost, and sex-equality cost. The authors consider a specific measure of performance: the balance cost, which is the maximum of the total dissatisfaction of the two sides. The problem has been shown to be NP-hard, and the authors provide a heuristic algorithm that finds the optimal stable matching in balance cost. Although the algorithm is not guaranteed to run in polynomial time, the authors provide empirical evidence that it runs efficiently on practically relevant instances, outperforming existing algorithms in terms of running time or solution quality.

**Strengths:**

1. The paper is remarkably well-written and easy to follow, and the background on the problem is well-explained.
2. The main algorithm is rigorously stated and analyzed, with intuitive graphical illustrations to aid understanding.
3. The empirical evaluation against existing algorithms is solid and convincing, showing that the proposed algorithm is competitive in terms of running time and solution quality.

**Weaknesses:**

I have no major concerns about this paper and am happy to see it accepted. However, I have a few minor comments that the authors may consider addressing.

1. **Relevance.** This paper is of high quality, and I personally appreciate the results. However, it seems to me that the results are mainly of interest to the discrete algorithms community. It would be helpful if the authors could discuss the paper’s relevance to the broader AI community and why ICLR would be the right venue for this paper.
2. **Experiment design.** Some aspects of the experiments could be improved. For example, when reading Figure 4(d), I was initially surprised that `ENUM-` performs worse on smaller instances than on larger ones. Then I realized that this is because you limited the number of iterations for `ENUM-`, and for some reason, the smaller instances required more iterations for `ENUM-`. However, the running time of `ENUM-` on smaller instances is still much shorter than on larger instances. From a practical perspective, I think it would be more informative to limit the running time of the algorithms instead of the number of iterations.

**Questions:**

1. In practical applications, what algorithm and performance measure are typically used for the stable marriage problem? Is the balance cost the dominant measure in practice? And does brute-force enumeration suffice in practice?
2. Have you considered how to construct instances where the proposed algorithm performs poorly? What are the limitations of the proposed algorithm?
3. Feel free to respond to the points raised in the "Weaknesses" section.

---

> ### Author Response · Authors · 2024-11-15
> **We construct and present hard instances in our experimental study**
>
> Thank you for the thorough review and the time invested in it.
>
> >discuss the paper’s relevance to the broader AI community and why ICLR would be the right venue for this paper.
>
> The AI and ML communities exhibit a continuous interest in stable matchings problems and their repercussions. Recent work [A] initiated the study of deep learning for the automated design of two-sided matching mechanisms, in order to understand tradeoffs between strategy-proofness and stability, which cannot be achieved simultaneously, training differentiable matching mechanisms that map discrete preferences to valid randomized matchings. We find that our work offers a viable alternative to such learning mechanisms, as it is independent of any mechanism among agents, and works directly in the space of rotations to find the exact optimal matching for balance cost. As machine learning pipelines are introduced in matching market design, we think it is vital to allow space for the publication of exact algorithms in machine learning venues, which provide a standard of comparison that learning-based algorithms will need to strive for.
>
> [A] Sai Srivatsa Ravindranath, Zhe Feng, Shira Li, Jonathan Ma, Scott Duke Kominers, David C. Parkes: Deep Learning for Two-Sided Matching. CoRR abs/2107.03427 (2021)
>
> >When reading Figure 4(d), I was initially surprised that ENUM- performs worse on smaller instances than on larger ones. Then I realized that this is because you limited the number of iterations for ENUM-, and for some reason, the smaller instances required more iterations for ENUM-. However, the running time of ENUM- on smaller instances is still much shorter than on larger instances. From a practical perspective, I think it would be more informative to limit the running time of the algorithms instead of the number of iterations.
>
> We limit the number of instances exaimined to $10^7$. Indeed ENUM- finds the optimal when given more time.
>
> >In practical applications, what algorithm and performance measure are typically used for the stable marriage problem? Is the balance cost the dominant measure in practice? And does brute-force enumeration suffice in practice?
>
> The Balance cost is a very attractive cost measure, yet it is avoided due to the NP-hardess of the problem.
>
> >Have you considered how to construct instances where the proposed algorithm performs poorly?
>
> We construct and present exactly such hard instances in our experimental study.
>
> >What are the limitations of the proposed algorithm?
>
> It may eventually encounter the exponential complexity bottleneck in hard problem instances, though that did not happen in our experiments.

---

> > ### Comment · Reviewer_fH5T · 2024-11-17
> >
> > Thank you for the response. It helped to clarify a few questions I had during the review.
> >
> > I will stand by my original positive recommendation.

---

### Official Review · Reviewer_kArv · 2024-10-30

**Soundness:** 2
**Presentation:** 2
**Contribution:** 2
**Rating:** 3
**Confidence:** 4

**Summary:**

This paper addresses the Balanced Stable Marriage (BSM) problem, which aims to achieve fair, stable matchings in two-sided markets by balancing dissatisfaction across both sides. The authors introduce ISORROPIA, which finds exact, balanced stable marriages by focusing on promising subsets within the solution space. Through experimentation, ISORROPIA outperforms existing approaches, particularly in time efficiency, without sacrificing the fairness objective.

**Strengths:**

The paper aims to achieve an exact solution to the Balanced Stable Marriage (BSM) problem, addressing fairness by minimizing dissatisfaction across both sides.  The introduced algorithm reduces computational overhead compared to baseline methods.

**Weaknesses:**

1. The paper lacks a comprehensive analysis of performance in diverse real-world applications.

2. The algorithm assumes static, pre-defined preference lists on both sides, which may not align with real-world two-sided markets where preferences are changing over time.

3.  The algorithm focuses on a subset of promising antichains to reduce computational costs. However, this selection could inadvertently introduce bias by excluding other potentially fair matchings that fall outside the selected subset.

4. The paper compares ISORROPIA primarily with traditional baselines and established heuristics. Including comparisons with more recent advancements or adaptive heuristics in stable matching (such as learning-based stable matching) could be more better.

**Questions:**

1. ISORROPIA narrows down the search space by focusing on promising antichains, yet the complexity of identifying these subsets is not thoroughly discussed. What is the theoretical bound on time complexity for extracting and evaluating these promising antichains, and under what conditions might this process become infeasible as the market size grows?

2.  Given that ISORROPIA minimizes the balance cost by searching over three specific sets of antichains, is there a mathematical guarantee that this focus always yields the exact minimum balance cost? If not, what bounds can be established for the deviation of ISORROPIA's solution from the true optimum?

3. What is the sensitivity of ISORROPIA’s performance to these initial stable marriages?

4. The rotation graph forms the basis for ISORROPIA’s local search, where each rotation represents a possible move in the solution space. How does the structure of the rotation graph (e.g., density of edges, distribution of antichains) affect the convergence and efficiency of ISORROPIA? Can we quantify the relationship between graph properties and the number of steps required to reach the optimal solution?

5. The paper states that the balance cost objective is non-convex, adding complexity to the optimization process. Can ISORROPIA guarantee convergence to a global minimum in all instances of the BSM problem, or are there instances where the algorithm might get stuck in local minima? If so, under what mathematical conditions does ISORROPIA ensure it escapes local minima?

---

> ### Author Response · Authors · 2024-11-15
> **Isorropia always finds the exact optimal solution**
>
> Thank you for the elaborate review.
>
> >W1: The paper lacks a comprehensive analysis of performance in diverse real-world applications.
>
> We do provide such analysis in our experimental study with diverse real-world data, including synthetic data, spatial data, non-spatial data, and hard instances.
>
> >W2: may not align with real-world two-sided markets where preferences are changing over time.
>
> Yes, dynamic preference lists pose an interesting direction for future work. In the current work, we contribute an algorithm that returns the **exact solution** to the NP-hard problem of minimizing the worst dissatisfaction on one side.
>
> >W3: exclud[es] other potentially fair matchings that fall outside the selected subset.\
> >Q2: is there a mathematical guarantee that this focus always yields the exact minimum balance cost?\
> >Q5: Can ISORROPIA guarantee convergence to a global minimum in all instances of the BSM problem?
>
> Yes, Isorropia finds the **exact global minimum balance cost** in all instances of the BSM problem, as our analysis in Section 4.3 establishes via Theorems 2, 3, and 4. The described bias does not arise, as matchings outside the selected subsets cannot yield an optimal solution to the BSM problem.
>
> >W4: [compare] with more recent advancements or adaptive heuristics in stable matching (such as learning-based stable matching).
>
> To our knowledge, such learning-based heuristics, such as [A], do not address the balanced stable matching problem, hence are not directly comparable to our solution.
>
> [A] Sai Srivatsa Ravindranath, Zhe Feng, Shira Li, Jonathan Ma, Scott Duke Kominers, David C. Parkes: Deep Learning for Two-Sided Matching. CoRR abs/2107.03427 (2021)
>
> >Q1: What is the theoretical bound on time complexity for extracting and evaluating these promising antichains?\
> >Q4: Can we quantify the relationship between graph properties and the number of steps required to reach the optimal solution?
>
> We discuss the complexity in Lines 438-441 and Appendix A.3 (line 903-913). As our analysis shows, complexity depends on the size $|R|$ of the rotation set and the depth of the rotation graph. In the worst case, the number of stable marriages $N$ can grow exponentially in the market size $n$, rendering all methods infeasible. Table 10 presents empirical statistics that illustrate the relationship between the number of rotations and the number of stable marriages.
>
> >Q3: What is the sensitivity of ISORROPIA’s performance to these initial stable marriages?
>
> These initial stable marriages do not alter the sets of promising antichains which Isorropia searches to find the exact optimal solution and which hence affect its performance; these sets depend on the rotation graph structure.

---

> ### Author Response · Authors · 2024-11-30
> **Look forward to your feedback**
>
> Dear Reviewers,
>
> As we approach the end of the rebuttal phase, we look forward to receiving your feedback. Please feel free to share any additional concerns you may have. If our response addresses your concerns and clarifies any potential misunderstandings, we would greatly appreciate it if you could reconsider the rating.

---

### Official Review · Reviewer_sQQa · 2024-11-08

**Soundness:** 3
**Presentation:** 2
**Contribution:** 3
**Rating:** 5
**Confidence:** 3

**Summary:**

This paper looks at the problem of stable matching in two-sided matching markets, which is a well-studied problem in the field of matching / mechanism design. In particular, the authors tackle the problem of finding a fair matching, as measured by comparing the two sides of the market, that is NP hard in its standard formulations: namely looking at the Balanced Stable Marriage problem. The balanced cost objective is non-convex and its minimization is NP hard. They are specifically motivated by the bottleneck created by the fact that the search space of all stable matching can grow exponentially in the size of the problem, and finding such a matching is infeasible in various real world applications. They authors present an algorithm, ISORROPIA, which uses a local search heuristic to reduce the search space and give an exact solution for instantiations that are reasonable in size. They do so by leveraging the structure of all feasible solutions to compactly represent them as a partially ordered sets. They present a theoretical analysis of this algorithm and experiments, using synthetic and real data, to show that ISORROPIA is more time and cost efficient than standard heuristics.

**Strengths:**

Two-sided matching is a well-studied and well-motivated problem, from both a theoretical and practical perspective. The version of it that they study, called the Balanced Stable Marriage problem, aims to tackle a discrepancy that exists in the standard formulation of this two-sided matching which is optimal for individuals on one side of the market and pessimal for the other side. The efficiency bottleneck they aim to address in this problem is a practical barrier and the insights they present about the structure of these antichains may also prove useful for future work. The problem presentation in this work is well-motivated and the authors also do a reasonable job giving the key takeaways upfront. The experiments they present performed better than I would have expected and the authors present these results both on larger instances of synthetic data and real data from various domains.

**Weaknesses:**

There are two key concerns with this work

(1) Specific to the results, while the first few sections of this paper were well-presented, the results are at times hard to decipher and follow along. It would be great if the authors revised the paper to provide intuition throughout the presentation of the algorithm. In particular, how are these anti-chains constructed and why does the pruning of the search space not seem to lead to significantly suboptimal solutions?

(2) Since one of the most compelling things about this  ISORROPIA is that it may be of practical use, it would be great if the authors included several more synthetic experiments to show how the performance degrades in size and complexity of the problem.

(3) Setting aside the above questions, it is not clear that ICLR is the right venue for this work as there doesn't seem to be any learning component.

**Questions:**

See weakness section above.

**Details Of Ethics Concerns:**

No ethics concerns

---

> ### Author Response · Authors · 2024-11-17
> **The optimal solution is guaranteed to lie within the non-pruned part of the search space**
>
> Thank you for the thorough review and the time invested in it.
>
> >W1: provide intuition throughout the presentation of the algorithm.
>
> We provide such intuition in Sections 4.2 and 4.3. We first divide the entire solution space into three sets (i.e., the blue, pink and grey parts in Figure 2, Line 289), and then prune these three sets (cf. dotted box in Figure 2). To make it more intuitive, we provide examples of the construction of these antichains in Table 4 and explain them in Lines 359-361 and Lines 385-386.
>
> Our algorithm is rooted in the structure of all solutions, which includes transformations among stable matchings, antichains, and closed subsets, based on the rotation structure. As readers may not be familiar with this background, we provide a comprehensive review of the preliminaries in Appendix A.1, including a toy example to illustrate the elimination process.
>
> >W1: why does the pruning of the search space not seem to lead to significantly suboptimal solutions?
>
> This pruning allows for finding the optimal solution, as that is guaranteed to lie within the non-pruned part of the search space. Our analysis in Section 4.3 establishes that via Theorems 2, 3, and 4, consistently with the graph in Figure 2.
>
> >W2: show how the performance degrades in size and complexity of the problem.
>
> We have included a wide array of experiments within the space allowed, which allow for conclusions on practical applicability to be made.
>
> >W3: it is not clear that ICLR is the right venue for this work as there doesn't seem to be any learning component.
>
> First, the AI and ML communities exhibit a continuous interest in stable matching problems and their repercussions [A, B, C, D], aiming to compute equitable and efficient solutions balancing disparate impacts in multi-agent systems. Besides, recent work [E] initiated the study of deep learning for the automated design of two-sided matching mechanisms, in order to understand tradeoffs between strategy-proofness and stability, which cannot be achieved simultaneously, training differentiable matching mechanisms that map discrete preferences to valid randomized matchings. We find that our work offers a viable alternative to such learning mechanisms, as it is independent of any mechanism among agents, and works directly in the space of rotations to find the exact optimal matching for balance cost. As machine learning pipelines are introduced in matching market design, we think it is vital to allow space for the publication of exact algorithms in machine learning venues, which provide a standard of comparison that learning-based algorithms will need to strive for.
>
> Further, matching problems have been adopted in learning models themselves. For example, the multi-object tracking model requires matching algorithms to solve the data association problem [F], while learning techniques in point cloud completion leverage a matching-based loss function [G].
>
> [A] Piotr Dworczak: Deferred acceptance with compensation chains. EC 2016
>
> [B] Nikolaos Tziavelis, Ioannis Giannakopoulos, Katerina Doka, Nectarios Koziris, and Panagiotis Karras: Equitable stable matchings in quadratic time. NeurIPS 2019
>
> [C] Nikolaos Tziavelis, Ioannis Giannakopoulos, Rune Quist Johansen, Katerina Doka, Nectarios Koziris, and Panagiotis Karras: Fair procedures for fair stable marriage outcomes. AAAI 2020
>
> [D] Sulian Le Bozec-Chiffoleau, Charles Prud'homme, Gilles Simonin: Polynomial Time Presolve Algorithms for Rotation-Based Models Solving the Robust Stable Matching Problem. IJCAI 2024
>
> [E] Sai Srivatsa Ravindranath, Zhe Feng, Shira Li, Jonathan Ma, Scott Duke Kominers, David C. Parkes: Deep Learning for Two-Sided Matching. CoRR abs/2107.03427 (2021)
>
> [F] Paul Voigtlaender, Michael Krause, Aljosa Osep, Jonathon Luiten, Berin Balachandar Gnana Sekar, Andreas Geiger, Bastian Leibe: MOTS: Multi-Object Tracking and Segmentation. CVPR 2019
>
> [G] Shunran Zhang, Xiubo Zhang, Tsz Nam Chan, Shenghui Zhang, Leong Hou U: A Computation-Aware Shape Loss Function for Point Cloud Completion. AAAI 2024

---

> ### Author Response · Authors · 2024-11-30
> **Look forward to your feedback**
>
> Dear Reviewers,
>
> As we approach the end of the rebuttal phase, we look forward to receiving your feedback. Please feel free to share any additional concerns you may have. If our response addresses your concerns and clarifies any potential misunderstandings, we would greatly appreciate it if you could reconsider the rating.

---

### Official Review · Reviewer_8Cf2 · 2024-11-09

**Soundness:** 3
**Presentation:** 2
**Contribution:** 2
**Rating:** 6
**Confidence:** 3

**Summary:**

This paper develops an exact algorithm using carefully designed heuristics to solve the balanced stable marriage problem which is known to be NP-hard and demonstrates that the proposed algorithm is faster in practice than existing exact algorithms. In a stable marriage instance, there are two disjoint subsets or sides of agents, e.g., men and women, with each side having ranked preferences over agents on the other side. The goal is to find a matching \mu, a collection of disjoint pairs of men and women, that is stable, i.e., no pair of man and women prefer each other than their partner in \mu. Here, the rank that an agent i has for their partner \mu(i) can be thought of as a proxy for their dissatisfaction with the matching \mu. The total dissatisfaction of a side is the sum of dissatisfactions of agents on that side.

In the balanced stable marriage problem (BSM), the goal is to find a stable marriage which minimizes the maximum among the dissatisfaction of the two sides, i.e., to balance the dissatisfaction of the two sides. This is motivated by a well known property of the set of all stable marriages that there is a man-optimal (which is also woman-pessimal) and a woman-optimal (which is also man-pessimal) stable marriage, and that the set of all stable marriages form a distributive lattice, and that it is possible to explore all stable marriages, from the man-optimal to woman-optimal stable matching through a series of operations called rotation elimination, each of which increases dissatisfaction of the men's side and decreases the dissatisfaction of the women's side. The set of rotations forms a poset and can be represented by a directed acyclic graph (DAG).

The proposed algorithm performs an intelligent search of the space of all stable matchings using heuristics and pruning strategies obtained from a careful analysis of the structure of stable matchings and a classification of the set of rotations and "anti-chains", sets of rotations with no shared predecessors in the DAG or rotations. The paper identifies that it is sufficient to explore three sets of anti-chains (and induced stable matchings) in order to find a balanced stable matching from which they identify two sets of candidate rotations. This is the main technical contribution of the paper. The proposed algorithm takes as input the DAG of rotations, computable in polynomial time, and produces a balanced stable matching as output. The algorithm works by performing a local search by enumeration of stable matchings due to a restricted set of rotations. The local search is potentially exponential.

The section on experiments validates their approach with the proposed algorithm performing better than a naive approach in practice in terms of running time. A comparison with approximation algorithms is also included showing to what extend the minimax BSM objective is achieved by various algorithms in comparison with the proposed exact algorithm.

**Strengths:**

- The proposed method is novel and interesting. BSM is an interesting and well motivated problem in the stable matchings research community with several potential applications. The paper is overall well written but has some typos which makes it confusing to read.
- In terms of running time proposed exact algorithm outperforms the naive enumeration approach and compares favorably with approximation algorithms.

**Weaknesses:**

- I do not find a major weakness except the fit within ICLR. My lower score for the contribution is due to the review criteria which clearly states "Are the results valuable to share with the broader ICLR community". I am not confident that they are.

**Questions:**

- Is there a typo in line 304?

---

> ### Author Response · Authors · 2024-11-15
> **Exact algorithms provide a standard that learning-based methods need to strive for**
>
> We appreciate your comprehensive review and the time invested to it.
>
> We understand the major concern is the fit within ICLR. The AI and ML communities exhibit a continuous interest in stable matchings problems and their repercussions. Recent work [A] initiated the study of deep learning for the automated design of two-sided matching mechanisms, in order to understand tradeoffs between strategy-proofness and stability, which cannot be achieved
> simultaneously, training differentiable matching mechanisms that map discrete preferences to valid randomized matchings. We find that our work offers a viable alternative to such learning mechanisms, as it is independent of any mechanism among agents, and works directly in the space of rotations to find the exact optimal matching for balance cost. As machine learning pipelines are introduced in matching market design, we think it is vital to allow space for the publication of exact algorithms in machine learning venues, which provide a standard of comparison that learning-based algorithms will need to strive for.
>
> [A] Sai Srivatsa Ravindranath, Zhe Feng, Shira Li, Jonathan Ma, Scott Duke Kominers, David C. Parkes: Deep Learning for Two-Sided Matching. CoRR abs/2107.03427 (2021)
>
> >Is there a typo in line 304?
>
> Yes, there was a typo in the subscript. Thank you for noticing.

---

### Meta-Review · Area_Chair_bjq9 · 2024-12-17

**Metareview:**

Reviewers generally liked the proposed method for finding balanced stable matchings and appreciated the informative empirical studies and good exposition. On the negative side, the biggest concern is the fitness of the paper to ICLR. After the rebuttal, reviewers still have this concern but also don't think this should not be the reason to immediately reject the paper. Some reviewers asked for more questions for the proposed method, which I think can be good questions for future work. Eventually, among the reviewers who participated in the discussions, the consensus is borderline accept.

**Additional Comments On Reviewer Discussion:**

Reviewer kArv and sQQa did not participate in the discussion. sQQa's review reads like a 6, and I think kArv's review is a bit harsh. The paper itself has reasonably good quality, so the main question is whether it is good enough to outweigh the fitness issue. Considering everything, accept (with the possibility of bumping down) was recommended.

---

### Decision · Program_Chairs · 2025-01-22

Accept (Poster)